# Excessive ammonium assimilation by plastidic glutamine synthetase causes ammonium toxicity in *Arabidopsis thaliana*

Takushi Hachiya [1,2,3✉], Jun Inaba[4], Mayumi Wakazaki[4], Mayuko Sato [4], Kiminori Toyooka [4], Atsuko Miyagi[5], Maki Kawai-Yamada[5], Daisuke Sugiura[2], Tsuyoshi Nakagawa[1], Takatoshi Kiba [2,4], Alain Gojon[6] & Hitoshi Sakakibara [2,4]

Plants use nitrate, ammonium, and organic nitrogen in the soil as nitrogen sources. Since the elevated $CO_2$ environment predicted for the near future will reduce nitrate utilization by $C_3$ species, ammonium is attracting great interest. However, abundant ammonium nutrition impairs growth, i.e., ammonium toxicity, the primary cause of which remains to be determined. Here, we show that ammonium assimilation by GLUTAMINE SYNTHETASE 2 (GLN2) localized in the plastid rather than ammonium accumulation is a primary cause for toxicity, which challenges the textbook knowledge. With exposure to toxic levels of ammonium, the shoot GLN2 reaction produced an abundance of protons within cells, thereby elevating shoot acidity and stimulating expression of acidic stress-responsive genes. Application of an alkaline ammonia solution to the ammonium medium efficiently alleviated the ammonium toxicity with a concomitant reduction in shoot acidity. Consequently, we conclude that a primary cause of ammonium toxicity is acidic stress.

[1] Department of Molecular and Functional Genomics, Interdisciplinary Center for Science Research, Shimane University, Matsue, Japan. [2] Graduate School of Bioagricultural Sciences, Nagoya University, Nagoya, Aichi, Japan. [3] Institute for Advanced Research, Nagoya University, Nagoya, Aichi, Japan. [4] RIKEN Center for Sustainable Resource Science, Yokohama, Kanagawa, Japan. [5] Graduate School of Science and Engineering, Saitama University, Saitama, Saitama, Japan. [6] Biochimie et Physiologie Moléculaire des Plantes, CNRS/INRA/SupAgro-M/Montpellier University, Montpellier, France. ✉email: takushi. hachiya@life.shimane-u.ac.jp

Nitrate and ammonium are the main sources of nitrogen (N) for most plants. Recent studies suggest that elevated $CO_2$ reduces nitrate reduction in $C_3$ species, such as wheat and *Arabidopsis*, whereas ammonium utilization does not decrease[1]. One estimate predicts that only a 1% drop in nitrogen use efficiency could increase worldwide cultivation costs for crops by about $1 billion annually[2]. Therefore, increasing ammonium use by crops is an important goal for agriculture as $CO_2$ levels rise in the world; however, millimolar concentrations of ammonium as the sole N source causes growth suppression and chlorosis in plants, compared with nitrate or lower concentrations of ammonium[3–5]. This phenomenon is widely known as ammonium toxicity, but the primary cause of impaired growth remains to be identified; in the present study, ammonium toxicity is defined as shoot growth suppression with toxic levels of ammonium as the sole N source compared with nitrate.

Plants grown in high ammonium conditions show several distinct characteristics from those grown in nitrate[3–5]. These toxic symptoms have evoked several hypotheses about the toxic causes, including futile transmembrane ammonium cycling, deficiencies in inorganic cations and organic acids, impaired hormonal homeostasis, disordered pH regulation, and the uncoupling of photophosphorylation; however, some of the symptoms are not directly associated with growth suppression by ammonium toxicity[6], making it difficult to determine the toxic cause. Several efforts have isolated ammonium-sensitive mutants in *Arabidopsis thaliana* and determined their causative genes[4,5]. *GMP1* is a causal gene whose deficiency causes stunted growth of primary roots under high ammonium conditions[7]. Given that GMP1 is crucial for synthesizing GDP-mannose as a substrate for *N*-glycosylation, a lack of *N*-glycoproteins could be involved in ammonium hypersensitivity. In accordance with this hypothesis, the ammonium-dependent inhibition of primary root growth was shown to be partly attenuated by the lack of a GDP-mannose pyrophosphohydrolase that hydrolyzes GDP-mannose to mannose 1-phosphate and GMP[8]. In another study, a genetic screen focusing on severely chlorotic *Arabidopsis* leaves identified *AMOS1*, a gene encoding a plastid metalloprotease, as a factor for improving ammonium tolerance[9]. Transcriptome analysis revealed that an AMOS1-dependent mechanism regulates more than half of the transcriptional changes triggered by toxic levels of ammonium. On the other hand, recent studies found that ammonium toxicity was partly alleviated by deficiencies in EIN2 and EIN3, regulators of ethylene responses, or by the application

of ethylene biosynthesis and action inhibitors[10,11]. This suggests that ammonium toxicity would be mediated via the ethylene signaling pathway.

The above-described genetic studies have succeeded in determining molecular components closely associated with ammonium toxicity. Nevertheless, the initial event that triggers ammonium toxicity remains to be identified and characterized. To address this question, we screened ammonium-insensitive *Arabidopsis* lines that were expected to attenuate the toxicity and isolated *ammonium-insensitive 2 (ami2)*. Interestingly, the defect in *ami2* was downregulation of the *GLUTAMINE SYNTHETASE 2* (*GLN2*) gene encoding an ammonium assimilatory enzyme localized in the plastid. We identified that in the presence of toxic levels of ammonium, large levels of proton production, due to excessive assimilation of ammonium by GLN2, aggravate the acidic burden and lead to plant toxicity.

## Results

**A genetic screen isolated an ammonium-insensitive mutant.** To find ammonium-insensitive lines, a gain-of-function population of the *Arabidopsis* FOX (full-length cDNA overexpressing) lines[12] was used. An apparent ammonium-insensitive mutant was identified that shows enhanced growth of cotyledons that are greener than wild-type (Col) when grown on 10 mM ammonium in the form of 5 mM $(NH_4)_2SO_4$ as the sole N source; the mutant was named *ami2* (Fig. 1a). The fresh weights of *ami2* 11-d-old shoots were approximately double those of Col when grown on ammonium (Fig. 1b). A similar growth enhancement in *ami2* was observed in the media containing 10 mM $NH_4Cl$ as the sole N source (Supplementary Fig. 1a, b), confirming that *ami2* is tolerant to ammonium. In the following experiments, we used 5 mM $(NH_4)_2SO_4$ as the ammonium source. In contrast, in media containing 10 mM nitrate or 5 mM ammonium plus 5 mM nitrate, the shoot fresh weights of *ami2* were less than those of Col. In media containing 10 mM ammonium, the percentage increase in fresh weight of *ami2* relative to Col was much larger for shoots (by ca. 110%) than for roots (by ca. 50%) (Fig. 1c). The greater shoot growth in *ami2* was reduced in media with lower concentrations of ammonium (0.4, 2 mM), in which the shoot growth of Col was greater than that when grown on media containing 10 mM ammonium (Fig. 1d). Moreover, nitrate addition in the presence of 10 mM ammonium attenuated the deficiency in shoot growth more effectively in Col than in *ami2*, decreasing the growth difference in a concentration-dependent

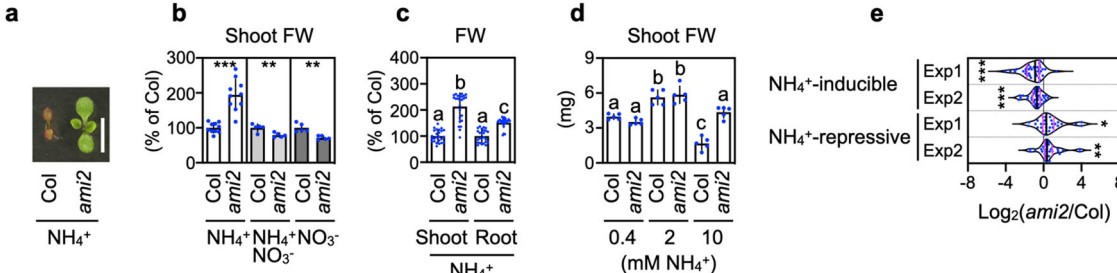

**Fig. 1 Enhanced shoot growth of *ami2* in the presence of 10 mM ammonium. a** A representative photograph of shoots from the 11-d-old wild-type (Col) and *ami2* grown on media containing 10 mM ammonium. The scale bar represents 5 mm. **b** Fresh weights (FW) of shoots from 11-d-old Col and *ami2* grown on media containing 10 mM ammonium (mean ± SD; $n = 10$), 5 mM ammonium nitrate (mean ± SD; $n = 5$), or 10 mM nitrate (mean ± SD; $n = 5$). **c** FW of shoots and roots from 11-d-old Col and *ami2* grown on media containing 10 mM ammonium (mean ± SD; $n = 26$). **d** FW of 11-d-old Col and *ami2* shoots grown on media containing 0.4, 2, or 10 mM ammonium (mean ± SD, $n = 5$). **e** Violin plots of the differences in the expression of the ammonium stress-responsive genes between the 10-d-old Col and *ami2* shoots 3 d after transfer to media containing 10 mM ammonium. The gene list was obtained from Li et al.[9] (for further details, see Supplementary Data 1). Two independent experiments (Exp1 and Exp2) were performed. Nine shoots from three plates constituted a single biological replicate. An individual violin plot shows the median (black line) and the 25th to 75th percentiles (magenta line). **b, d** Six shoots from one plate constituted a single biological replicate. **b, e** *$P < 0.05$; **$P < 0.01$; ***$P < 0.001$ (Welch's *t*-test). **c, d** Different letters denote significant differences at $P < 0.05$ (Tukey–Kramer's multiple comparison test).

manner (Supplementary Fig. 1c, d). A time-course analysis of shoot growth revealed that increased ammonium tolerance of the *ami2* plants compared to Col was significant as soon as 5 d after culture initiation (Supplementary Fig. 1e). These results indicate that ammonium tolerance in *ami2* shoots is manifested specifically under harsh ammonium conditions.

To corroborate this enhanced ammonium tolerance in *ami2*, we performed microarray experiments and compared the expression of genes responsive to toxic levels of ammonium[9] between the Col and *ami2* shoots growing in media containing 10 mM ammonium (Fig. 1e and Supplementary Data 1). The transcript levels of ammonium-inducible genes were significantly reduced in *ami2* shoots compared with Col shoots, whereas those of ammonium-repressive genes showed the opposite trend. A reverse transcription-quantitative PCR (RT-qPCR) analysis confirmed that expression of *CMCU*, *MIOX2*, and *PDH2*, ammonium-inducible genes[9,13], was upregulated in Col shoots under 10 mM ammonium compared with 10 mM nitrate, and this ammonium induction was significantly suppressed in *ami2* shoots (Supplementary Fig. 2a). We confirmed that the expression of a housekeeping gene *TIP41* was little changed, and that of nitrate-inducible marker gene *NIR*[14] was increased under 10 mM nitrate condition (Supplementary Fig. 2b, c). Collectively, these results indicate that ammonium toxicity is attenuated in *ami2* shoots.

**GLUTAMINE SYNTHETASE 2 is a causative gene for ammonium toxicity**. Next, to identify the causative gene in *ami2*, we recovered the transgene in a vector using specific primers and sequenced the construct. The gene was identified as *GLUTAMINE SYNTHETASE 2* (*GLN2*), the sole plastidic isoform in *A. thaliana* (Fig. 2a and Supplementary Fig. 3a). Because the transgene was driven by the cauliflower mosaic virus *35S* promoter, we expected that overexpression of *GLN2* would enhance ammonium tolerance; however, in media containing 10 mM ammonium, the transcript levels of *GLN2* in *ami2* shoots were downregulated to about 5% of those in Col (Fig. 2b). In contrast, among the major cytosolic *GLUTAMINE SYNTHETASE* genes (*GLN1s*), *GLN1;1* was upregulated in *ami2* shoots, but *GLN1;2* and *GLN1;3* were slightly downregulated (Supplementary Fig. 3b). Also, an immunoblot analysis using anti-GLN antibodies[15] confirmed that the protein levels of GLN2 were remarkably lower in *ami2* shoots compared with Col, whereas the signal intensities corresponding to GLN1s were comparable between the mutant and wild type (Fig. 2c). These findings suggested that overexpression of *GLN2* cDNA would result in a cosuppression event[16], which makes it difficult to test for phenotypic complementation by introducing the *GLN2* transgene. To ensure that reduced expression of *GLN2* enhances ammonium tolerance, we obtained another *GLN2*-deficient line having a T-DNA insertion at the 3′-UTR region of *GLN2* (SALK_051953, designated as *gln2*, Supplementary Fig. 3a). As expected, *gln2* phenocopied *ami2* in terms of the reduced *GLN2* and GLN2 expression (Fig. 2b, c), the enhanced ammonium tolerance (Fig. 2d, e and Supplementary Fig. 3c, d), and the lowered induction of ammonium-inducible genes when grown on ammonium (Supplementary Fig. 2a). Furthermore, a similar ammonium tolerance was observed in the GLN2-knockout line (SALK_071292, designated as *gln2-2*, Supplementary Fig. 3a, e, f) which was recently isolated in another research group[17]. Thus, we concluded that *GLN2* is a causative gene for ammonium toxicity.

**Shoot GLN2 causes while root GLN1;2 attenuates ammonium toxicity**. Previous studies had reported that mutants deficient in *AtGLN1;2* were hypersensitive to millimolar concentrations of

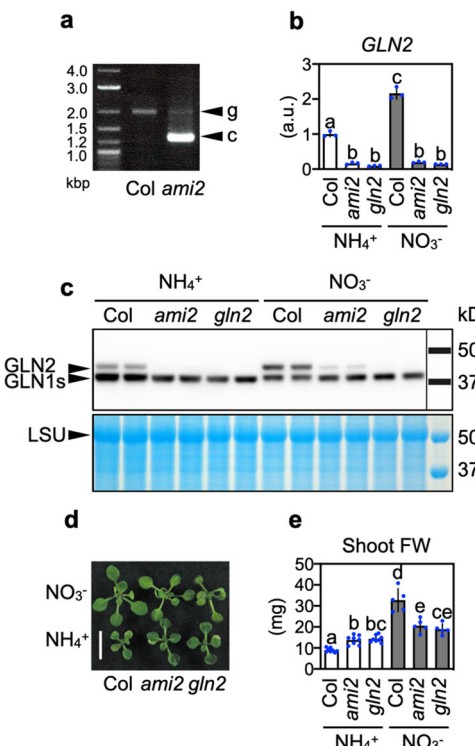

**Fig. 2 Downregulation of GLN2 enhances ammonium tolerance. a** Genomic PCR using *GLN2*-specific primers. g and c denote the PCR fragments derived from genomic DNA and cDNA sequences corresponding to *GLN2*, respectively. **b** Relative transcript levels of *GLN2* in the shoots of 10-d-old Col, *ami2*, and *gln2* 3 d after transfer to media containing 10 mM ammonium or 10 mM nitrate (mean ± SD; n = 3). Six shoots from two plates constituted a single biological replicate. **c** Immunodetection of GLN1s and GLN2 isoproteins using specific antisera following SDS-PAGE and immunoblotting of total proteins from the shoots of 12-d-old Col, *ami2*, and *gln2* 5 d after transfer to media containing 10 mM ammonium or 10 mM nitrate. LSU denotes large subunits of RuBisCO. **d** A representative photograph of shoots from 14-d-old Col, *ami2*, and *gln2* 7 d after transfer to media containing 10 mM ammonium or 10 mM nitrate. The scale bar represents 10 mm. **e** FW of shoots from 14-d-old Col, *ami2*, and *gln2* 7 d after transfer to media containing 10 mM ammonium (mean ± SD; n = 8) or 10 mM nitrate (mean ± SD; n = 5). Mean values of three shoots from one plate constituted a single biological replicate. **b**, **e** Different letters denote significant differences at P < 0.05 (Tukey–Kramer's multiple comparison test).

ammonium[18–20]. We also confirmed the ammonium hypersensitivity of *gln1;2-1* and *gln1;2-2* (Supplementary Fig. 4a–d). *GLN2* and *GLN1;2*, therefore, have opposite effects on ammonium toxicity. To discover how GLN2 and GLN1;2 are involved in the toxicity, we evaluated the distribution of *GLN2* and *GLN1;2* expressions between shoots and roots in Col plants. In the presence of ammonium or nitrate, the steady-state levels of *GLN2* expression were consistently higher in the shoots than in the roots, whereas expression of *GLN1;2* was much higher in the roots (Fig. 3a, b and Supplementary Fig. 5a, b), implying that both shoot GLN2 and root GLN1;2 could affect ammonium toxicity. To support this hypothesis, we performed a growth analysis using reciprocally grafted plants between Col and *ami2* (Fig. 3c and Supplementary Fig. 5c) and between Col and *gln1;2-1* (Fig. 3d and Supplementary Fig. 5d). Prior to the analysis, we confirmed that shoot expression of *GLN2* was lower in the *ami2*-derived shoots irrespective of root-genotype (Supplementary Fig. 6) because *GLN2* mRNA is suggested to be root-to-shoot mobile[21]. Only

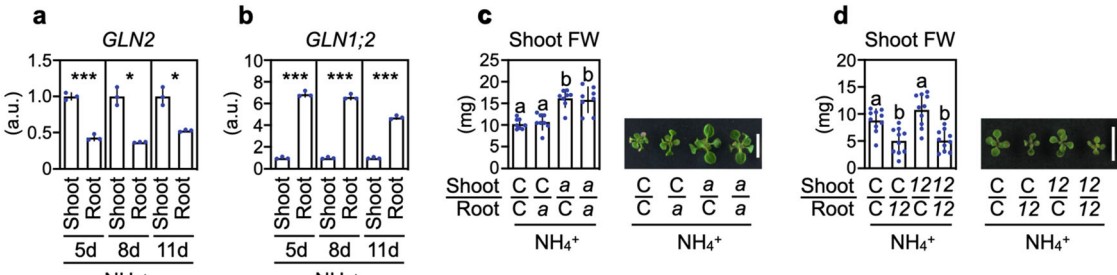

**Fig. 3 Shoot GLN2 causes ammonium toxicity, whilst root GLN1;2 attenuates it. a** Relative transcript levels of *GLN2* in the shoots and roots of 5-, 8-, or 11-d-old Col grown on media containing 10 mM ammonium (mean ± SD; *n* = 3). **b** Relative transcript levels of *GLN1;2* in the shoots and roots of 5-, 8-, or 11-d-old Col grown on media containing 10 mM ammonium (mean ± SD; *n* = 3). **a**, **b** Twelve shoots and roots from one plate constituted a single biological replicate. *$P < 0.05$; **$P < 0.01$; ***$P < 0.001$ (Welch's *t*-test). **c** FW of shoots from 17-d-old reciprocally grafted plants between Col (C) and *ami2* (*a*) 7 d after transfer to media containing 10 mM ammonium (mean ± SD; *n* = 8). **d** FW of shoots from 17-d-old reciprocally grafted plants between Col (C) and *gln1;2-1* 7 d after transfer to media containing 10 mM ammonium (mean ± SD; *n* = 10). **c**, **d** One shoot from one plate constituted a single biological replicate. Different letters denote significant differences at $P < 0.05$ (Tukey–Kramer's multiple comparison test). Representative photograph of 17-d-old shoots 7 d after transfer to media containing 10 mM ammonium are shown. The scale bar represents 10 mm.

when the scion was derived from *ami2* was shoot growth significantly enhanced in the presence of 10 mM ammonium (Fig. 3c). On the other hand, deficiency in root *GLN1;2* content was sufficient to decrease shoot growth in ammonium (Fig. 3d). Further, we observed that in ammonium-grown plants, the total enzymatic activities of GLNs were significantly reduced by ca. 30–40% in 5-d-old shoots of *ami2* and *gln2* and by ca. 40–60% in 5-d-old roots of *gln1;2-1* and *gln1;2-2* compared with Col (Supplementary Fig. 7a, b). Additionally, partially compensatory inductions of other *GLNs* were found in the mutants (Supplementary Fig. 7c, d). Our findings demonstrate that although shoot GLN2 causes ammonium toxicity in the shoot, root GLN1;2 attenuates ammonium toxicity.

**Lowered GLN2 activity reduces the conversion of ammonium to amino acids.** It is generally held that ammonium per se is a toxic compound[22]. On the other hand, a deficiency in *GLN2* content should lead to ammonium accumulation in the shoot. Our determination of shoot ammonium content revealed that *ami2* and *gln2* shoots grown on 10 mM ammonium both accumulated more than 100 μmol g$^{-1}$ fresh weight of ammonium (Fig. 4a), albeit the two mutants accumulated more fresh weight than Col (Supplementary Fig. 8). In Col shoots, the ammonium concentrations were 12.9 ± 0.9 μmol g$^{-1}$ and 2.2 ± 0.4 μmol g$^{-1}$ (mean ± SD) under 10 mM ammonium and 10 mM nitrate, respectively (Fig. 4a). This result indicates that ammonium assimilation by GLN2, rather than ammonium accumulation, triggers ammonium toxicity in the shoot.

An ample supply of ammonium increases the concentrations of amino acids compared with nitrate supply alone[6,23]. In particular, the molar ratios of Gln to Glu are elevated at higher ammonium levels, suggesting that Gln synthesis by glutamine synthetase (GLN) overflows glutamate synthase (GOGAT) capacity. Our hierarchical cluster analysis of amino acid content in shoots clearly demonstrated that the type of N source, i.e., 10 mM ammonium or nitrate, was the strongest determinant for plant amino acid composition (Fig. 4b and Supplementary Fig. 9a). In this analysis, Col and the *GLN2*-deficient lines are categorized into separate clusters depending on the N source. The molar ratio of Gln to Glu (Fig. 4c), total amino acid-N content per amino acid (Supplementary Fig. 9b), total amino acid-N content per fresh weight (Supplementary Fig. 9c), and the molar ratios of N to C in total amino acids (Supplementary Fig. 9d) were consistently larger in ammonium-grown shoots than nitrate-grown shoots, and this large ammonium-N input was partly but significantly

attenuated by *GLN2* deficiency. These findings suggest that the GLN2 reaction leads to excessive incorporation of ammonium-N into amino acids in shoots when toxic levels of ammonium are present. In addition, we observed that total N and protein concentrations were generally larger in ammonium-grown shoots than nitrate-grown shoots (Supplementary Fig. 10a, b). In ammonium-grown shoots, the total N concentrations were significantly reduced by *GLN2* deficiency (Supplementary Fig. 10a), and the protein concentrations were marginally decreased (Supplementary Fig. 10b). Collectively, the shoot GLN2 would facilitate the accumulation of assimilated N in the shoot under toxic ammonium condition.

**Ammonium assimilation by GLN2 causes acidic stress.** The amino acid profiles suggested that metabolic imbalances due to excessive ammonium assimilation by GLN2 could be a cause of ammonium toxicity. This hypothesis is supported by the observation that shoot growth of Col was much lower under 10 mM ammonium than under 2 mM ammonium and the shoot growth enhancement by *GLN2* deficiency was determined only under 10 mM ammonium as already explained above. We have previously demonstrated that nitrate addition at adequate concentrations mitigates ammonium toxicity without reducing amino acid accumulation[6]. Therefore, a phenomenon triggered by some GLN2-mediated process other than amino acid accumulation should be a cause of ammonium toxicity. Notably, the GLN reaction is a proton-producing process[24]. The stoichiometry of this reaction is two protons per each glutamine produced, one proton of which is derived from ATP hydrolysis and the other is from deprotonation of $NH_4^+$. Conversely, the subsequent ferredoxin-dependent glutamate synthase (Fd-GOGAT) reaction consumes two protons per one glutamine incorporated. Given that the molar ratio of Gln to Glu was about 10 in the ammonium condition but close to 1 in the nitrate condition (Fig. 4c), proton production in the ammonium condition could proceed beyond its consumption. Strikingly, a previous study found that 43% of ammonium-inducible genes correspond to acidic stress-inducible genes in *Arabidopsis* roots[13,25]. We found that ammonium-inducible genes whose expression was reduced by *GLN2* deficiency under 10 mM ammonium condition (Supplementary Fig. 2a) are known as acid-inducible genes[25]. Thus, we hypothesized that excessive ammonium assimilation by GLN2 causes acidic stress to the plants growing in ammonium.

We re-surveyed our microarray data by focusing on previously identified acidic stress-responsive genes[25] (Fig. 5a and

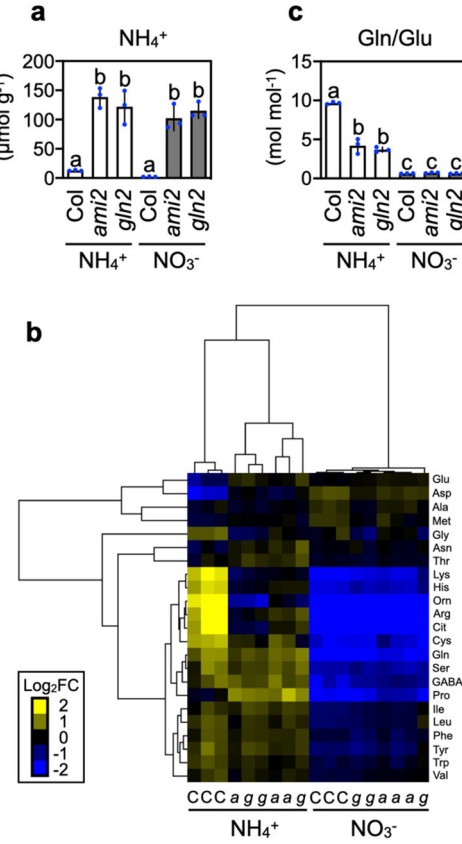

**Fig. 4 Lowered activity of GLN2 reduces the conversion of ammonium to amino acids. a** The shoot ammonium content of 12-d-old Col, *ami2*, and *gln2* 5 d after transfer to media containing 10 mM ammonium or 10 mM nitrate (mean ± SD; $n = 3$). Three shoots from one plate constituted a single biological replicate. **b** Hierarchical clustering of the shoot amino acid content of 12-d-old Col (C), *ami2* (*a*), and *gln2* (*g*) 5 d after transfer to media containing 10 mM ammonium or 10 mM nitrate. The color spectrum from yellow to blue corresponds to the relative content of each amino acid. The fold-change (FC) value on each amino acid was calculated by dividing the amino acid content of a sample by the mean content of all 18 samples. Log$_2$FC denotes the base-2 logarithm of FC. **c** The molar ratio of Gln to Glu in the shoots of 12-d-old Col, *ami2*, and *gln2* 5 d after transfer to media containing 10 mM ammonium or 10 mM nitrate (mean ± SD; $n = 3$). **a**, **c** Different letters denote significant differences at $P < 0.05$ (Tukey–Kramer's multiple comparison test). **b**, **c** Six shoots from two plates constituted a single biological replicate. Three biological replicates were sampled separately three times.

Supplementary Data 2). All acidic stress-inducible genes were entirely downregulated in *ami2* shoots compared with Col, whereas the acidic stress-repressive genes showed the opposite trend. The transcript levels of *ALMT1*, a typical acidic stress-inducible gene, were determined in the shoots of Col and *ami2* plants incubated in 10 mM ammonium or nitrate with or without methionine sulfoximine (MSX), an inhibitor of the GLN reaction (Fig. 5b). *ALMT1* expression was much higher in the ammonium-treated Col shoots than in the nitrate-treated samples. This ammonium-dependent induction was significantly diminished in the *ami2* shoots and was mimicked by MSX treatment. Also, other proton-inducible genes such as *GABA-T*, *GAD1*, *GDH2*, *PGIP1*, and *PGIP2*[26] were ammonium-inducible, and their inductions were suppressed or attenuated by *GLN2* deficiency (Supplementary Fig. 11a). GAD1 catalyzes the conversion from Glu to GABA with consumption of proton, acting as a pH-regulating pathway[26]. The molar ratios of GABA to Glu were larger in ammonium-grown shoots than nitrate-grown shoots, and these larger ratios were significantly reduced by *GLN2* deficiency (Supplementary Fig. 11b), implying facilitated proton consumption by GAD1 in a GLN2-dependent manner under ammonium condition. These results support our hypothesis associating ammonium assimilation with acidic stress. Moreover, in Col and *ami2* reciprocally grafted plants growing in the ammonium condition, *ALMT1* expression was significantly lower in the *ami2*-derived shoots than the Col-derived shoots (Supplementary Fig. 11c), indicating that shoot GLN2 locally causes acidic stress to the shoot. Furthermore, *ALMT1* expression was analyzed using grafted plants between Col and a mutant lacking the STOP1 transcription factor (*stop1-KO*) that induces *ALMT1* to respond to acidic stress[26] (Supplementary Fig. 11d). In the *stop1-KO*-derived shoots, the ammonium-dependent induction of *ALMT1* disappeared, reconfirming the notion that acidic stress occurs in plants growing in ammonium.

It is widely accepted that the reduction from nitrate to ammonium consumes a proton, suggesting that nitrate reduction could attenuate acidic stress caused by excess ammonium and might explain why nitrate addition alleviates ammonium toxicity. To verify this hypothesis, we analyzed shoot expression of *ALMT1* using grafted plants between Col and the *NITRATE REDUCTASE*-null mutant (designated as NR-null)[14] (Supplementary Fig. 11e). The addition of 2.5 mM nitrate diminished the ammonium-dependent *ALMT1* induction in the Col-derived shoots but not in the NR-null-derived shoots, thereby supporting the above hypothesis.

To obtain direct evidence for ammonium-dependent proton production, we measured the proton concentrations of water extracts from the Col and *ami2* shoots incubated in media containing 10 mM ammonium or nitrate with or without MSX (Fig. 5c). The ammonium-treated Col shoots contained the highest concentrations of protons; proton content was significantly decreased by *GLN2* deficiency and by MSX treatment to levels comparable to those in nitrate-treated shoots. A similar trend was observed among the Col, *ami2*, and *gln2* shoots grown on ammonium- or nitrate-containing media (Supplementary Fig. 11f).

The presence of ammonium in cultures generally acidifies the external media[13]. Thus, we quantified the proton efflux from the Col and *ami2* shoots incubated in media containing 10 mM ammonium or nitrate with or without MSX (Fig. 5d). Incubation of the Col shoots in the presence of ammonium strongly acidified the external media, which was alleviated by *GLN2* deficiency and by MSX treatment. A similar tendency was observed by qualitative measurements with a pH indicator of proton effluxes from mesophyll cells where GLN2 is predominantly expressed (Supplementary Fig. 11g). Thus, we conclude that ammonium assimilation by GLN2 without nitrate increases shoot acidity.

**Ammonium toxicity is closely associated with acidic stress**. If acidic stress rather than ammonium accumulation has a dominant effect on ammonium toxicity, an application of alkaline ammonia should reduce the toxicity. Given that the GLN2 reaction is a primary cause of increased acidic stress, an elevation in medium pH may increase the shoot growth of Col more effectively than that of the *GLN2*-deficient mutants. As expected, the addition of an ammonia solution to media containing 10 mM ammonium elevated the pH from 5.7 to 6.7 and significantly improved shoot growth with a concomitant decrease in acidity (Fig. 6a, b). Fresh weights of Col shoots grown at pH 6.7 increased by ca. 180% compared with those grown at pH 5.7, whereas fresh weights of *ami2* and *gln2* shoots only increased by ca. 30% and 60%, respectively (Fig. 6c). In addition, the acid-

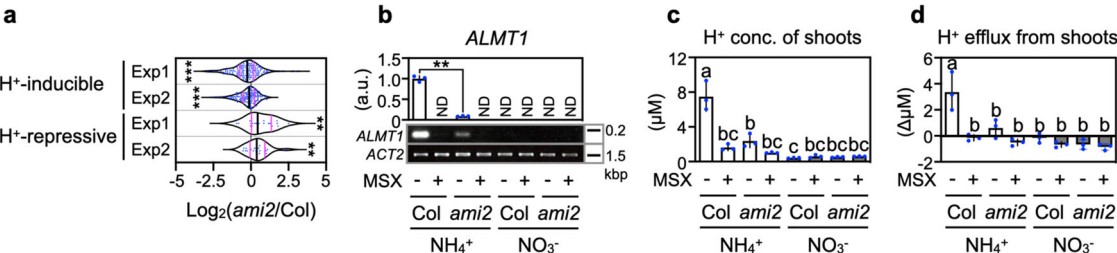

**Fig. 5 Ammonium assimilation by GLN2 causes acidic stress. a** Violin plots of the differences in expression of the acidic stress-responsive genes between the 10-d-old Col and *ami2* shoots 3 d after transfer to media containing 10 mM ammonium. The gene list was obtained from Lager et al.[25] (For further details, see Supplementary Data 2). Two independent experiments (Exp1 and Exp2) were performed. Nine shoots from three plates constituted a single biological replicate. An individual violin plot shows the median (black line) and the 25th to 75th percentiles (magenta line). **b** Effects of MSX treatment on the relative transcript level of *ALMT1* in the 10-d-old Col and *ami2* shoots 3 d after transfer to media containing 10 mM ammonium or 10 mM nitrate. The transcript levels were evaluated both by RT-qPCR (mean ± SD; $n = 3$) and semi-quantitative RT-PCR with agarose gel electrophoresis. *ACTIN2* (*ACT2*) was the internal standard. Three shoots from one plate constituted a single biological replicate. **c** Effects of MSX treatment on proton concentrations in water extracts from the 10-d-old Col and *ami2* shoots 3 d after transfer to media containing 10 mM ammonium or 10 mM nitrate (mean ± SD; $n = 3$). Three shoots from one plate constituted a single biological replicate. **d** Effects of MSX treatment on proton efflux rates from the 10-d-old Col and *ami2* shoots 3 d after transfer to media containing 10 mM ammonium or 10 mM nitrate (mean ± SD; $n = 3$). Three shoots from one plate constituted a single biological replicate. **a, b** *$P < 0.05$; **$P < 0.01$; ***$P < 0.001$ (Welch's $t$-test). **c, d** Different letters denote significant differences at $P < 0.05$ (Tukey–Kramer's multiple comparison test).

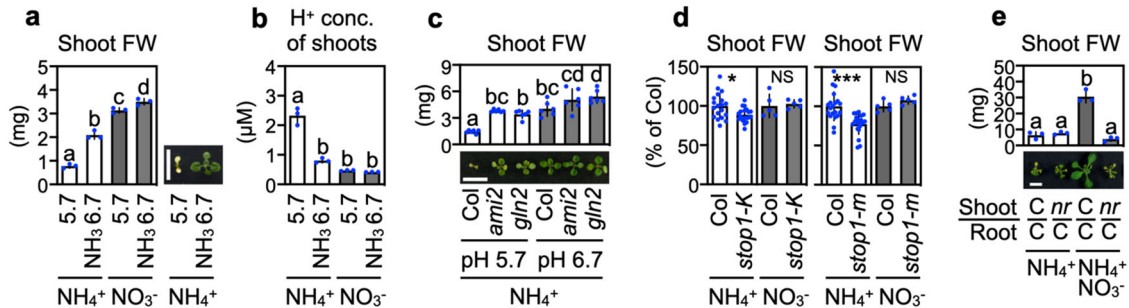

**Fig. 6 Ammonium toxicity is closely linked with acidic stress. a, b** Effects of $NH_3$ application on shoot FW and proton concentrations in water extracts of 5-d-old Col grown on media containing 10 mM ammonium or 10 mM nitrate (mean ± SD; $n = 3$). Thirty-seven shoots from one plate constituted a single biological replicate. The pH was adjusted to pH 5.7 with 1 N KOH; subsequently, a commercial 25% (v/v) ammonia solution was added to adjust the pH from 5.7 to 6.7. A representative photograph of 11-d-old shoots grown on 10 mM ammonium (12 plants per plate) is shown. **c** Effects of higher pH on the FW of shoots from 11-d-old Col, *ami2*, and *gln2* grown on media containing 10 mM ammonium for 11 d (mean ± SD; $n = 6$). Three shoots from one plate constituted a single biological replicate. The pH was adjusted to pH 5.7 with 1 N KOH; subsequently, 1 N NaOH was used to adjust the pH from 5.7 to 6.7 to maintain the potassium concentration constant among all samples. A representative photograph of 11-d-old shoots is shown. **d** FW of shoots from 11-d-old Col, *stop1-KO* mutant (*stop1-k*), and the *stop1* mutant (*stop1-m*) grown on media containing 10 mM ammonium (mean ± SD; $n = 20$) or 10 mM nitrate (mean ± SD; $n = 5$) for 11 d. Six shoots from one plate constituted a single biological replicate. *$P < 0.05$; **$P < 0.01$; ***$P < 0.001$ (Welch's $t$-test). **e** FW of shoots from 17-d-old plants grafted between Col (C) and the NR-null mutant (*nr*) 7 d after transfer to media containing 10 mM ammonium ($NH_4^+$) or 2.5 mM nitrate and 10 mM ammonium ($NH_4^+NO_3^-$) conditions (mean ± SD; $n = 3$). One shoot from one plate constituted a single biological replicate. A representative photograph of 17-d-old shoots 7 d after transfer to media is shown. **a–c, e** Different letters denote significant differences at $P < 0.05$ (Tukey–Kramer's multiple comparison test). The scale bar represents 10 mm.

sensitive *STOP1*-deficient mutants had slightly but significantly lower shoot growth when grown in 10 mM ammonium (Fig. 6d), although their acid-hypersensitivity has been described only in roots to date[26,27]. Moreover, the NR-null-derived shoots that lack a proton-consuming nitrate reduction capacity failed to attenuate ammonium toxicity by nitrate addition (Fig. 6e). Collectively, our results lead to the conclusion that acidic stress is one of the primary causes of ammonium toxicity.

**GLN2 causes ammonium toxicity independently of NRT1.1.** We have already reported that *Arabidopsis* NITRATE TRANS-PORTER 1.1 (NRT1.1), acting as a nitrate transceptor, also aggravates ammonium toxicity[28]. This result was recently confirmed by another group[11]. Thus, we investigated whether NRT1.1 and GLN2 increase the sensitivity to ammonium through a common mechanism. A RT-qPCR analysis revealed that deficiency of either *NRT1.1* or *GLN2* did not downregulate the

expression of the other gene (Supplementary Fig. 12a), and that *GLN2* expression was almost 3-times higher in *nrt1.1* than in Col. Therefore, the enhanced ammonium tolerance of *nrt1.1* cannot be explained by reduced *GLN2* expression as in *gln2*. Moreover, a homozygous double mutant of *NRT1.1* and *GLN2* showed slightly but significantly larger shoot fresh weight, leaf number, shoot diameter, and chlorophyll content compared with any of the single mutants (Supplementary Fig. 12b–d). These findings suggest that *NRT1.1* and *GLN2* would be implicated in ammonium-sensitivity independently.

## Discussion

Although ammonium is believed to be a toxic compound for plant growth, our results demonstrate that ammonium assimilation by shoot GLN2 rather than ammonium accumulation is a major cause of ammonium toxicity. In plants growing in toxic levels of ammonium as a sole N source, assimilation of

ammonium by GLN2 would occur largely due to bypassing nitrate reduction as the rate-limiting step for N assimilation. The resultant increase in the ratio of Gln to Glu content (Fig. 4c) corresponds to the preferential enhancement of the proton-producing GLN reaction over the proton-consuming GOGAT reaction. The metabolic imbalances exerted by the GLN2 reaction and GLN2-related pathways could lead to the production of large amounts of protons in shoot cells that stimulate proton effluxes to the apoplasm; however, the volume of the shoot apoplasm is probably too small to accommodate such a large proton efflux. Thus, in the presence of toxic levels of ammonium, the GLN2 reaction causes acidic stress inside and outside the cells and triggers acidic stress responses that modulate gene expression (Fig. 5 and Supplementary Fig. 11). Given that the wild-type when grown at a higher pH phenocopies the ammonium-insensitive lines at lower pH, the acidic stress-sensitive mutants show ammonium-hypersensitivity, and proton-consuming nitrate reduction alleviates ammonium toxicity (Fig. 6), we conclude that acidic stress is one of the primary causes for ammonium toxicity in *A. thaliana*. In this framework, upregulation of *Arabidopsis* *NIA1* and *NIA2* genes encoding nitrate reductase (NR)[25,26] and activation of spinach NR[29] responding to acidic stress are understandable regulatory responses in the context of maintaining cellular pH homeostasis.

The present study does not address how ammonium-dependent acidification triggers growth deficiency at the cellular and subcellular scales. The chloroplastic localization of GLN2 indicates that proton production must occur within chloroplasts in the elevated ammonium condition. A previous study reported abnormal chloroplast membrane structure including swollen compartments at late stages of ammonium toxicity[30]; however, we did not find any similar structural changes in the shoots of ammonium-grown plants (Supplementary Fig. 13a), where the intermediates of the Calvin-Benson cycle were not depleted compared with nitrate-grown shoots (Supplementary Fig. 13b). These observations do not support a deficiency in chloroplast function as a primary cause of ammonium toxicity. Apoplastic pH in sunflower leaves and cytosolic pH in carrot cell suspensions decrease after application of millimolar levels of ammonium[31,32]. The ammonium-inducible genes whose expression is down-regulated by *GLN2* deficiency, *PGIP1* and *PGIP2* (Supplementary Fig. 11a), contribute to cell wall stabilization under acidic stress[27], implying apoplastic acidification as a target of ammonium toxicity. On the other hand, in the presence of toxic levels of ammonium, the expression of GABA shunt-related genes (Supplementary Fig. 11a), the molar ratio of GABA to Glu (Supplementary Fig. 11b), and oxygen uptake rates[33] are increased as biochemical pH-stats[26,34] that may represent an intracellular acidic burden. Given that changes in pH environments influence a wide spectrum of physiological processes, elucidating the relationship between ammonium-dependent acidification and growth deficiency awaits future study.

At the whole-plant scale, our grafting work demonstrated that root GLN1;2 activity attenuates ammonium toxicity in the shoots, whilst shoot GLN2 activity causes the condition (Fig. 3). Considering that GLN1;2 is the ammonium-inducible low-affinity enzyme expressed in the epidermis and cortex of roots, and its deficiency elevates ammonium levels in xylem sap when ammonium is supplied[20], root GLN1;2 could act as a barrier to prevent the shoot-to-root transport of ammonium, thus avoiding ammonium assimilation by shoot GLN2. In oilseed rape plants, replacing 3 mM nitrate in a nutrient solution with 10 mM ammonium increased the ammonium levels in xylem sap linearly with time, attaining concentrations >5 mM[35], which could indicate breaking through the barrier. On the other hand, we observed that shoot expression of *GLN1;2* and *GLN1;3* encoding

low-affinity GLN1 isozymes was significantly larger under 10 mM ammonium than 10 mM nitrate (Supplementary Fig. 3b). Moreover, a larger protein signal corresponding to shoot GLN1s was detected when plants received ammonium rather than nitrate nutrition (Fig. 2c and Supplementary Fig. 14a, b), implying the above-mentioned barrier function of GLN1;2 and GLN1;3 in the shoot. Whilst a large activity of shoot GLN1s could reduce ammonium assimilation by shoot GLN2, shoot GLN1s reaction per se may produce an excessive proton, thereby causing ammonium toxicity. Thus, we compared the proton concentrations of water extracts from the Col, *ami2*, and *gln1;2 gln1;3* shoots incubated in media containing 10 mM ammonium (Supplementary Fig. 14c). Interestingly, we found that double knockout of *GLN1;2* and *GLN1;3* did not significantly reduce the proton concentrations, whereas it decreased shoot GS activity by ca. 35%, being comparable to *GLN2* deficiency (Supplementary Fig. 14c, d). This suggests that shoot GLN1;2 and GLN1;3 could assimilate excessive ammonium without causing proton accumulation in the shoot, supporting their barrier function. It is a challenging issue to address how GLN1s and GLN2 have different effects on pH regulation in the future.

It is widely believed that the plastidic isoform of GS (GS2) plays a central role in re-assimilating photorespiratory ammonium. The mutants lacking GS2 in barley cannot survive under photorespiratory air conditions[36]. In *Lotus japonicus*, the leaves of GS2-deficient mutants show severe necrotic phenotype when transferred from high to low $CO_2$ condition[37]. On the other hand, a recent study reported that *Arabidopsis* GS2-deficient mutants can complete their life cycle under air conditions, albeit smaller and slightly chlorotic than the wild-type[17]. Even under conditions that lower photorespiration, the *GLN2*-deficient mutants show growth impairment, suggesting an important role of GLN2 in primary nitrogen assimilation. Also, we did not find a strong suppression of vegetative growth of *ami2* and *gln2* plants grown in the soil under photorespiratory conditions compared with the wild-type (Supplementary Fig. 15a), although their seedling growth during 1 week after imbibition in the half-strength Murashige and Skoog media was significantly retarded (Supplementary Fig. 15b, c). Moreover, *ami2* and *gln2* plants propagated the seeds whose total N and C concentrations and germination rates were comparable to those of the wild-type (Supplementary Fig. 15d–f). Thus, in *A. thaliana*, GLN2, GLN1s, and GDHs proteins could redundantly contribute to the re-assimilation of photorespiratory ammonia. This may be why no mutant lacking *GLN2* has been isolated as a photorespiratory mutant from *Arabidopsis* plants. On the other hand, the present study did not clarify how improved ammonium insensitivity in *Arabidopsis GLN2*-deficient mutants is related to photorespiration. To elucidate this, it awaits many efforts in comparison between photorespiratory and non-photorespiratory conditions.

The present study demonstrated that *GLN2* and *NRT1.1* reduce ammonium tolerance via separate mechanisms when plants experience high ammonium conditions (Supplementary Fig. 12). By contrast, these genes are nitrate-inducible genes that are crucial for plant adaptation to nitrate-dominant environments[38,39]. This observation suggests that the adaptive traits to nitrate and ammonium could be exclusive, and therefore, breeding ammonium-tolerant crops might sacrifice their adaptability to nitrate. *Arabidopsis* growth is enhanced by elevating $CO_2$ more remarkably with ammonium as the sole N source than with nitrate, although, even under elevated $CO_2$, the absolute value of plant biomass is still greater when receiving nitrate[1]. Overcoming ammonium toxicity could lead to elevated $CO_2$-adapted crops in terms of their mode of N utilization.

## Methods

**Plant materials and growth conditions**. *A. thaliana*, accession Columbia (Col) was the control line used in this study. The T-DNA insertion mutants *gln2* (SALK_051953), *gln2-2* (SALK_071292)[17], *gln1;2-1* (SALK_145235)[19], *gln1;2-2* (SALK_102291)[18], *stop1-KO* (SALK_114108)[26], and *nrt1.1* (SALK_097431)[28] were purchased from the European Arabidopsis Stock Centre. The *Arabidopsis* FOX lines[12] and the ethyl methanesulfonate-mutagenized *stop1* mutant (psi00011)[26] were obtained from the RIKEN BioResource Research Center. Seeds of the NR-null mutant[14] and *gln1;2 gln1;3* mutant[20] were acquired from Dr. Nigel M. Crawford (University of California, San Diego) and Dr. Soichi Kojima (Tohoku University, Sendai), respectively.

Seeds were surface-sterilized and sown in plastic Petri dishes (diameter 90 mm, depth 20 mm, Iwaki, Tokyo, Japan) containing about 30-mL N-modified Murashige and Skoog medium, supplemented with 10 mM MES, 2% (w/v) sucrose, and 0.25% (w/v) gellan gum (pH 5.7). The N sources were 10 mM $KNO_3$ (10 mM nitrate condition), 5 mM $NH_4NO_3$ with 10 mM KCl (5 mM ammonium nitrate condition), or 5 mM $(NH_4)_2SO_4$ with 10 mM KCl (10 mM ammonium condition). The seeds were placed in the dark at 4 °C for 3 d. Plants were grown in a horizontal position under a photosynthetic photon flux density of 100–130 µmol $m^{-2}$ $s^{-1}$ (16 h light/8 h dark cycle) at 23 °C. For the transfer experiments, surface-sterilized seeds were sown in plastic Petri dishes (length, 140 mm; width, 100 mm; depth, 20 mm; Eiken Chemical Co. Ltd., Taito-ku, Tokyo, Japan) containing 50-mL half-strength N-modified Murashige and Skoog medium containing 2.5 mM ammonium as the sole N source, supplemented with 4.7 mM MES, 1% (w/v) sucrose, and 0.4% (w/v) gellan gum (pH 6.7)[40]. This culture condition allowed almost uniform growth of different lines, alleviating ammonium toxicity. The seeds were placed in the dark at 4 °C for 3 d. The plants were grown in a vertical position for 7 d under a photosynthetic photon flux density of 100–130 µmol $m^{-2}$ $s^{-1}$ (16 h light/8 h dark cycle) at 23 °C. The plants were transferred with sterilized tweezers to varying N conditions and grown in a horizontal position for further experiments. Other details of plantlet cultivation are presented in the Results section and the figure legends.

**Isolation of ammonium-insensitive lines**. Forty-five seedlings per plate of *Arabidopsis* FOX lines[12] were grown on the above-mentioned N-modified Murashige and Skoog media containing 5 mM $(NH_4)_2SO_4$ with 10 mM KCl as the sole N source for 11 d in a horizontal position. The ammonium-insensitive candidate lines were visually selected based on shoot size and color of the cotyledons or true leaves. Then, the next generation of each candidate line was obtained to refresh the seeds. In the second screen, the ammonium insensitivity of the 10 mM ammonium-grown 11-d-old line was evaluated quantitatively by determining their seedling fresh weights in comparison with the wild-types (more than 10 biological replicates). The lines having statistically significant increases in the fresh weight compared with the wild-types were regarded as the ammonium-insensitive ones. In total, seed mixtures that included 15,800 lines (pss10001-10016) were used for a series of screens, resulting in the isolation of three lines (*ami1-ami3*). The transgene included in the candidate line was recovered in a vector (pCR8/GW/TOPO, ThermoFisher Scientific, Waltham, MA, USA) using specific primers[12] and identified by sequencing. The expression of the native and transgene was checked by RT-PCR and qRT-PCR. The identified genes for *ami1* and *ami3* were different from those for *ami2*.

**Extraction of total RNA**. Shoots and roots were harvested, immediately frozen with liquid $N_2$, and stored at −80 °C until use. Frozen samples were ground with a Multi-Beads Shocker (Yasui Kikai Corp., Osaka Prefecture, Osaka, Japan) using zirconia beads (diameter, 5 mm). Total RNA was extracted using the RNeasy Plant Mini Kit (Qiagen) and on-column DNase digestion according to the manufacturer's instructions.

**Reverse transcription and quantitative (real-time) PCR**. Reverse transcription was performed using a ReverTra Ace qPCR RT Master Mix with gDNA Remover (Toyobo Co. Ltd., Tokyo, Japan) according to the manufacturer's instructions. The synthesized cDNA was diluted 10-fold with distilled water and used in quantitative PCR (qPCR). Transcript levels were measured using a StepOnePlus Real-Time PCR System (ThermoFisher Scientific). The obtained cDNA (2 µL) was amplified in the presence of 10-µL KAPA SYBR FAST qPCR Kit (Nippon Genetics Co. Ltd., Tokyo, Japan), 0.4-µL specific primers (0.2 µM final concentration), and 7.2-µL sterile water. Transcript levels were quantified using a relative standard curve with *ACTIN3* as the internal standard. A dilution series of total cDNAs were used as templates to generate the standard curves. The primer sequences used in the experiments are shown in Supplementary Data 3.

**Immunoblot analysis of GLN isoproteins**. Immunoblot analysis of GLN proteins was performed based on a previously described method[40]. Shoots were harvested, frozen with liquid $N_2$, and stored at −80 °C until use. Frozen samples were ground with a Multi-Beads Shocker (Yasui Kikai Corp.) using zirconia beads (diameter, 5 mm). Total proteins were extracted with 10 volumes of sample buffer [2% (w/v) SDS, 62.5 mM Tris–HCl (pH 6.8), 7.5% (v/v) glycerol, 50 mM DTT and 0.01% (w/v) bromophenol blue] containing a protease inhibitor tablet (Roche Diagnostics,

Basel, Switzerland). The extracts were incubated at 95 °C for 5 min followed by cooling on ice and centrifugation at 20,400 × *g* at room temperature (20–25 °C) for 10 min. A 10-µL aliquot of the supernatant (equivalent to approximately 1-mg fresh sample) was subjected to SDS-PAGE in a 12% (w/v) gel (TGX FastCast Acrylamide Kit, Bio-Rad Laboratories, Hercules, CA, USA) and transferred to a PVDF membrane (Trans-Blot Turbo Mini PVDF Transfer Packs, Bio-Rad Laboratories, Hercules, CA, USA) using HIGH MW with a semi-dry blotting system (Trans-Blot Turbo Transfer System, Bio-Rad Laboratories). The membrane was incubated overnight in blocking buffer containing 5% (w/v) ECL Prime Blocking Agent (GE Healthcare, Little Chalfont, UK), 0.1% (v/v) Tween-20, 50 mM Tris–HCl, and 150 mM NaCl (pH 7.6). The blocked membrane was then incubated for 1 h with a 1/10,000 dilution of the polyclonal antibodies raised against maize cytosolic glutamine synthetase[15] that was obtained from Dr. Hitoshi Sakakibara (Nagoya University, Nagoya, Aichi, Japan). After rinsing with a buffer containing 0.1% Tween-20, 50 mM Tris–HCl, and 150 mM NaCl (pH 7.6), the antigen–antibody complex was detected using a 1/100,000 dilution of horseradish peroxidase-conjugated to goat anti-rabbit IgG (NA935, GE Healthcare) and visualized by chemiluminescent detection (ECL Prime, GE Healthcare) using ImageQuant LAS 3000 mini (Fujifilm, Tokyo, Japan). The signal intensities for each band corresponding to GLN1s and GLN2 isoproteins were quantified using Image J software, version 10.2. After detection, the membrane was stained with the GelCode Blue Stain Reagent (ThermoFisher Scientific).

**Microarray analysis**. Total RNA was extracted as described above. RNA quality was assessed using an Agilent 2100 bioanalyzer (Agilent Technologies). RNA amplification, labeling, hybridization, and scanning with the 3′ IVT Express Kit (Affymetrix) and the GeneChip Arabidopsis Genome ATH1 Array (Affymetrix) were conducted according to the manufacturer's instructions. The data set from the microarray chips was normalized by the Microarray Suite 5.0 (MAS5) method (Affymetrix). When the signal detection of a transcript was labeled "Absent" or "Marginal", the sequence was omitted from the subsequent quantitative analysis. The raw data are shown in Supplementary Data 4. Changes in respective gene expression levels between Col and *ami2* were represented as logarithms to base 2 of the ratios in signal intensities.

**Grafting between shoots and roots**. *Arabidopsis* seedlings were grown for 4 d on media containing 2.5 mM ammonium as the sole N source at pH 6.7[40] before grafting. Each seedling was perpendicularly cut at the hypocotyl with the tip of an injection needle (NN-2613S, TERUMO, Tokyo, Japan) on a mixed cellulose membrane (HAWP09000, MERCK MILLIPORE, Darmstadt, Germany). The scion was kept in touch with the rootstock through a section of 0.4 mm diameter silicon tubing as described[41]. The grafted plants were grown for another 4 d at 27 °C, and then, for 2 d at 23 °C. Plants without adventitious roots were transferred to the fresh media and further grown and used for analyses of growth and gene expression.

**Determination of glutamine synthetase activity**. GLN activity was determined as the ADP-dependent conversion rate of L-glutamine to γ-glutamylhydroxamate[42]. Shoots and roots were harvested, immediately frozen with liquid $N_2$, and stored at −80 °C until use. Frozen samples were ground with a TissueLyser II (QIAGEN) using 5 mm zirconia beads. The powder was mixed with 10 volumes of extraction buffer (100 mM Tris–HCl, pH 7.5, 1% (w/v) PVP-40, 1 mM EDTA, 1 mM $MnCl_2$, 0.5% (v/v) β-mercaptoethanol, 0.1 mM 4-APMSF). The extracts were centrifuged at 12,000 × *g* at 4 °C for 10 min. The reaction was started by adding 45 µl of pre-incubated assay buffer (40 mM imidazole-HCl, pH 7.0, 20 mM sodium arsenate, 0.5 mM ADP, 3 mM $MnCl_2$, 60 mM $NH_2OH$, 30 mM L-glutamine) to 5 µL of the supernatant. The mixture was incubated at 30 °C for 15 min. The reaction was stopped by adding 30 µl of $FeCl_3$-TCA-HCl solution (2.6% $FeCl_3 \cdot 6H_2O$, 4% tri-chloroacetic acid in 1 N HCl). Ferric γ-glutamylhydroxamic acid was measured by spectrophotometric absorbance at 540 nm.

**Determination of amino acids and photosynthetic intermediates**. Amino acids and photosynthetic intermediates were extracted and their concentrations were determined based on the method reported by Miyagi et al. (2010)[43] with minor modifications. Shoots were harvested, frozen with liquid $N_2$, and stored at −80 °C until use. Frozen samples were ground with a Multi-Beads Shocker (Yasui Kikai Corp.) using zirconia beads (diameter, 5 mm). Metabolites were extracted with 50% (v/v) methanol containing 50 µM PIPES and 50 µM methionine sulfone (internal standard). After the first centrifugation (21,500 × *g*, 5 min, 4 °C), the supernatant was transferred to a 3 kDa cut-off filter (UFC500324, Millipore, Billerica, MA, USA) and centrifuged again (13,700 × *g*, 30 min, 4 °C). The filtered extract was used for capillary electrophoresis-mass spectrometry (CE-MS) analysis.

Measurements of metabolites were performed using an Agilent 6400 series triple quadrupole CE-MS system (CE; 7100 Capillary Electrophoresis, MS; 6420 Triple Quad LC/MS, Agilent Technologies, Santa Clara, CA, USA) in multi reaction monitoring (MRM) mode[43,44]. A DB-WAX capillary (polyethylene glycol-coated, 100 cm × 50 µm i.d., Agilent Technology) with 20 mM ammonium acetate (pH 8.5) as the running buffer was used for measuring anionic compounds (e.g., organic acids and phosphorylated compounds), and an uncoated fused silica

capillary (100 cm × 50 μm i.d., GL Sciences, Tokyo, Japan) with 1 M formic acetate was used for cationic compounds (e.g., amino acids). MS analysis at the applied −25 kV for anions was carried out in negative ion mode. Cations were determined in positive ion mode (applied voltage 25 kV). For MS stabilization, 5 mM ammonium acetate (for anions) or 0.1% formic acid (v/v) (for cations) in 50% (v/v) methanol was used as sheath solution, applied to the capillary at 10 μL min$^{-1}$ using an isocratic HPLC pump (Agilent 1200 series) equipped with a 1:100 splitter. The capillary voltage (± 3500 V) and the drying nitrogen gas (at 320 °C) flow (8 L min$^{-1}$) were held constant for approximately 30 min during each electrophoresis run. Quantitative accuracy was determined using known concentrations of standard reference compounds using Agilent MassHunter Software, version B.07.00.

**Determination of ammonium**. Ammonium was extracted and its concentration was determined with slight modifications to the method reported by Hachiya et al.[42]. Shoots were harvested, immediately frozen with liquid $N_2$, and stored at −80 °C until use. Frozen samples were ground with a Multi-Beads Shocker (Yasui Kikai Corp.) using zirconia beads (diameter, 5 mm). One mL of 0.1 N HCl and 500 μL of chloroform were added to the frozen powder, followed by vortexing for 15 min. The mixture was centrifuged at 12,000 × g at 8 °C for 10 min. The aqueous phase was transferred to a microtube containing 50 mg of acid-washed activated charcoal (No. 035-18081; Wako, Osaka, Japan). The mixture was vortexed and centrifuged at 20,400 × g at 8 °C for 10 min. The ammonium content of the supernatant was spectroscopically determined using an Ammonia Test Kit (No. 277-14401, Wako) according to the manufacturer's instructions.

**Determination of H$^+$ concentration in a water extract of shoots**. For Fig. 5c and Supplementary Fig. 14c, plants grown on media containing 10 mM ammonium or 10 mM nitrate were submerged in 5 mL of the corresponding liquid medium without MES buffer in the presence or absence of 1 mM methionine sulfoximine (MSX). The plants were returned to their original positions on each medium and incubated for 5 h in the light. Shoots were harvested, immediately frozen with liquid $N_2$, and stored at −80 °C until use. For Fig. 6a and Supplementary Fig. 11f, the shoots grown on media containing 10 mM ammonium or 10 mM nitrate were harvested without prior incubation in liquid media, immediately frozen with liquid $N_2$, and stored at −80 °C until use. Ten volumes of $H_2O$ for Fig. 5c and Supplementary Figs. 11f, 14c and 40 volumes of $H_2O$ for Fig. 6a were added to the frozen powder followed by centrifugation at 10,000 × g at room temperature for 10 min. The pH of the supernatant was measured with a portable pH meter (B212, Horiba, Ltd.). The values were converted to proton concentrations.

**Determination of H$^+$ efflux from shoots**. Excised shoots from the plants grown on media containing 10 mM ammonium or 10 mM nitrate were submerged in 100 volumes of the corresponding liquid medium without MES buffer in the presence or absence of 1 mM methionine sulfoximine (MSX). The submerged shoots were incubated in liquid media for 5 h in the light. The pH changes in the incubation media were measured with a portable pH meter (B212, Horiba, Ltd.). The changes in pH were converted to changes in proton concentrations.

**Determination of H$^+$ efflux from mesophyll protoplasts**. Mesophyll protoplasts were prepared with slight modifications to the method reported by Endo et al.[45]. In one dish, 20 plants were grown for 7 d on media containing 2.5 mM ammonium as the sole N source at pH 6.7[40] in a vertical position. Ten plants were transferred to fresh medium with the same composition and grown for another 7 d. Two plants were further transferred to a medium containing 10 mM ammonium and grown for 3 d in a horizontal position. The adaxial epidermal surface was affixed to plastic tape, whereas the abaxial epidermal surface was attached to a strip of 15-mm wide Scotch transparent tape (3 M, St. Paul, MN, USA). The Scotch tape was then carefully pulled away from the plastic tape, peeling away the abaxial epidermal surface cell layer. The plastic tape samples with adhering peeled leaves were transferred to 1.5 mL microtubes containing 1 mL of an enzyme solution [0.75% (w/v) cellulase "Onozuka" R10 (Yakult, Tokyo, Japan), 0.25% (w/v) macerozyme "Onozuka" R10 (Yakult), 0.4 M mannitol, 8 mM CaCl₂, and 5 mM MES, pH 5.6]. The tube was slowly rotated for 20 min at room temperature. The liberated protoplasts were filtered through a 30-μm nylon filter (NY30-HD, SEMITEC, Osaka, Japan), and collected by centrifugation at 100 × g at 4 °C for 5 min. The protoplast suspensions of leaf samples from 16 plants were pooled. The supernatant was removed, and the protoplasts were resuspended in 1 mL of a wash solution [0.4 M mannitol and 8 mM CaCl₂, pH 6.5] followed by centrifugation at 100 × g at 4 °C for 5 min. This washing step was repeated once more, after which the protoplasts were suspended in 100 μL of the wash solution. Ten μL of protoplast suspension was mixed with 90 μL of an assay solution [0.4 M mannitol, 8 mM CaCl₂, 0.02% bromocresol purple (BCP) pH 6.5] supplemented with 5 mM (NH₄)₂SO₄ and 10 mM KCl in the presence or absence of 1 mM MSX, 10 mM KCl, or 10 mM KNO₃. The mixture was incubated at 23 °C in the light until color changes were observed.

**Observation of chloroplast ultrastructure in mesophyll cells with TEM**. For ultrastructural analyses, true leaves from Col, *ami2*, and *gln2* 5 d after the transfer

to media containing 10 mM ammonium or 10 mM nitrate were analyzed by transmission electron microscopy according to a previously described method[46] with modification. Leaves were fixed with 4% (w/v) paraformaldehyde and 2% (v/v) glutaraldehyde in 50 mM sodium cacodylate buffer (pH 7.4) overnight at 4 °C. After the samples were post-fixed with 1% (w/v) osmium tetroxide for 3 h and dehydrated in a graded methanol series, the samples were embedded in EPON812 resin. Micrographs were recorded using a JEM-1400 (JEOL) transmission electron microscope.

**Statistical and cluster analyses**. All statistical analyses were conducted using R software, version 3.6.2. Two-sided tests were used for Welch's *t*-test. Adjustments were made for Tukey–Kramer's multiple comparison test. Hierarchical clustering was performed using Gene Cluster software, version 3.0, with "Correlation (uncentered)" as the similarity metric and "Average linkage" as the clustering method. The results were visualized using Java TreeView software, version 1.1.6r4. Other details of analyses are provided in the "Results" section and in the table and figure legends.

## Data availability

The raw microarray data used in this study are available in the ArrayExpress database at EMBL-EBI under accession number E-MTAB-10796. Source data are provided with this paper.

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

## Acknowledgements

Seeds of the *A thaliana* NR-null mutant and *gln1;2 gln1;3* were kindly provided by Dr. Nigel M. Crawford (University of California, San Diego) and by Dr. Soichi Kojima (Tohoku University, Sendai), respectively. This work was supported by the Building of Consortia for the Development of Human Resources in Science and Technology, by the Japan Society for the Promotion of Science KAKENHI Grant No. JP17K15237, by the Program for Promoting the Enhancement of Research Universities, Nagoya University by the Inamori Foundation, by the Agropolis Foundation No. 1502-405, by a Grant-in-Aid for Young Scientists from Shimane University, and by the joint research fund from Yanmar Holdings Co., Ltd. FOX lines were developed by the Plant Genome Project of RIKEN PSC, and were provided by the RIKEN BRC through the National BioResource Project of the MEXT/AMED, Japan.

## Author contributions

T.H. conceived and initiated the project. T.H., A.G., and H.S. designed the experiments. T.H., J.I., M.W., M.S., K.T., A.M., M.K.-Y., D.S., and T.K. performed the experiments and data analysis. T.H., T.N., A.G., and H.S. wrote the manuscript.

## Competing interests

The authors declare no competing interests.
