## [Peer Review File · Nature Communications]

Reviewers' comments:

Reviewer #1 (Remarks to the Author):

Excessive ammonium assimilation by plastidic glutamine synthetase causes ammonium toxicity in *Arabidopsis thaliana*.

Screening *Arabidopsis* FOX lines under 10 mM toxic ammonium, authors identified over-expressing lines with higher tolerance to ammonium. Although the *ami2* line identified carries the 35S:GLN2 construct, this mutant was in fact defective for GLN2 probably due to silencing, and it was accordingly phenocopied by the *gln2* mutant.

Then using elegant grafting assays, authors demonstrated that mutating GLN2 led to higher ammonium in shoots but lower tissue acidification, and then better tolerance to toxic ammonium levels. The fact that ammonium toxicity was due to acidification of shoot tissue was then verified using several approaches that combined grafting with other mutants and proton measurements.

In addition, the study shows the antagonistic role of GLN2 in shoot and GLN1;2 in root regarding ammonium toxicity, and that *Nrt1.1* and GLN2 reduce ammonium tolerance by two different ways. There are many reports about ammonium toxicity and several mentioned acidification as one of the reason of the toxicity. The present study provides an elegant demonstration for that, and also reveals the surprising role of GLN2 in the tolerance to high ammonium. Indeed, one would have rather imagined that higher GLN2 activity leads to better tolerance to ammonium. This indeed challenge the textbook knowledge

This manuscript of high quality of science is also very well written and the results clearly presented are new and highly interesting for all the plant physiologists and molecular biologists interested in plant nutrition and nutrient signalling.

I will then have only few minor comments:

- From the paper of Rubio-Asensio, J.S. & Bloom, A.J., I would not say that high CO₂ inhibits nitrate utilization but reduce nitrate utilization, and it should also be said that even under high CO₂, nitrate utilization remains better for plant growth than ammonium utilization.

- It was described in barley (Kendall et al., 1986; Wallsgrove et al., 1987) and in *Lotus japonicas* (Orea et al., 2002) that GS2 mutants have photorespiratory symptoms when transferred from high CO₂ to air. Their hypersensitivity to air atmosphere was explained as due to lack of reassimilation of photorespiratory ammonium.

Is there any phenotype of *gln2* mutant under nitrate conditions compared to WT? Maybe not on plates but at later stage of development in greenhouse? No role in photorespiration that necessitate high CO₂ for survival? This might be discussed.

- In Fig.4b the hierarchy clustering presents metabolite fold changes. FC means comparison of two conditions or two genotypes. Please explain in the legend what are exactly these FC values.

- In Fig.11a, it looks that chloroplast structure of *gln2* and *ami2* is different from WT under ammonium

conditions. Could you please comment?

Reviewer #2 (Remarks to the Author):

The article from Hachiya et al. aimed to identify the genes responsible for ammonium toxicity with full length overexpression lines. Surprisingly, a causal gene was glutamine synthetase 2, which was well characterized in barley. The physiological function of GS2 was defined in the assimilation of the ammonia during photorespiration in previous work. This work is ambitious because it challenges the textbook knowledge. However, the relation between GS2 and photorespiration was not discussed in this work. It is surprising because it is the textbook knowledge.

This point is critical.

I found some problems in data presentation and writing in this work. I hope my comments are valuable and helpful for this work to increase its value.

The manuscript concludes that ammonium toxicity is acidic stress. It has been already known. Therefore, the novelty of this work should be that GS2 was a causative gene for acidic stress. I think this work showed well GS2 dependent proton production. However, the mechanism was still not clear. The manuscript explained that GS2 reaction release two proton in its reaction. I am not sure this point. Is there some other examples showing the strong impact on huge proton production by specific enzymatic reaction?

According to the logic from the manuscript, fd-gogat mutants should be more sensitive to ammonium, and overexpression of Fd-GOGAT should be more tolerant to ammonium. I would suggest to probe them.

Ammonium supply changes the balance between GS2 and GS1. GS1 seems a major form in shoot in higher ammonium condition. Does GS1 contribute to proton production?

Since most of the data in this manuscript was derived from the two gs2 mutants. However, unfortunately both of them are mysterious. One was a overexpression line with suppression, the other was T-DNA insertion in 3'-UTR. Therefore, it is essential to discuss complementation lines. Recent publication shows successful GS2 overexpression in rice. Therefore, I suppose overexpression of GLN2 cDNA would always not result in co-suppression. Or, there is famous gs2 mutants in barley. It would be nice to test ammonium toxicity in barley mutant.

This work discuss small seedlings in whole work. Since they are very small, the data may reflect the characteristics of seeds. Are the seed germination speed between Col and mutants same? Are the quality and quantity of nitrogen in Col and mutants similar?

This work often mixes different culture condition. It is confusing. Fig.1a and Fig.2d shows very different plant growth. The size of plant in Fig. 2d was much bigger than that in Fig. 1a. In Fig. 2d, plants were grown in pH 6.7 and were transferred to pH 5.7. The reason of pH switching was finally explained in

Figure 6. The structure of data presentation is not easy to follow. In addition, phenotype was surely significant, but was not obvious in Fig. 2d. Even inside of Fig. 2 there was difference among b, c, and d. Plant growth and response to the ammonium should be very different in different culture condition as the manuscript claimed in introduction.

This work shows fresh weight in whole work. It must be dry weight. Mutants might have much higher moisture than wild-type, and might be diluting ammonium.

This work often shows ratio instead of absolute value. It masks some valuable information. It was difficult to understand the plant size difference between Fig. 1 and Fig. 2. It was not possible to compare the growth between Fig. 2d and Fig. 3c.

Fig. 2e, the data between ammonium and nitrate should be compared. Other Figures, 4a, 4c, 2b and Fig. 2e were different data interpretation.

Fig. 4a, why ammonium concentration in ammonium grown plant and nitrate grown plant are similar?

Fig. 2, 3, 4 and 5, comparison between low ammonium and high ammonium should be discussed.

L24 Plant can use organic nitrogen.

L61, there are several definitions of ammonium toxicity in different ammonium concentration in previous works.

This work should compare previous definitions and define its ammonium toxicity in the manuscript.

L375-377, the isolation of mutants should be explained in detail.

How are they isolated, and how are they confirmed as mutants?

L392-393, this medium contains MES and sucrose?

The medium used in this work was not clear. Why MES was not able to buffer the acidification in this work?

Reviewer #3 (Remarks to the Author):

This work presents interesting data on the glutamine synthetase-2 (GS2) assimilation reaction as responsible, through the H⁺ release, of an acidic stress which would be subjacent to the stress by ammonium nutrition. The authors provide a radical new idea on the underlying mechanisms of stress by ammonium, which is studied only in the plant Arabidopsis. The work includes an extensive set of data around the biochemical mechanisms, the gene expression and metabolite induction, which support their hypothesis. The data are also supported by an important set of supplementary figures and a table. In general, the data seems well presented although some questions need to be further clarified.

- The role of GS2 in Arabidopsis during photorespiration may be the most difficult point to accept from

this new hypothesis. It is considered among plant biologist that GS2 has an essential role on the photorespiration during recycling of the photorespiratory ammonium and in the assimilation of ammonium from reduction of nitrate at the plastids. Thus, the barley mutants on GS2 needed to be grown on 1% oxygen to avoid growth limitation during photorespiratory conditions (Wallsgrave et al., 1987). In the *Lotus japonicum* GS2 mutants, the plants needed to be cultivated at least at 4000 ppm of CO₂, while 7000 ppm of CO₂ were regularly used to avoid photorespiratory conditions (Perez-Delgado et al., 2015). In this work, the GS2 expression is 95% lost in *ami2* mutant according to the authors, and beyond that in GSN mutant. This should at least induce a phenotype of the *ami2* and GSN plants under nitrate but nothing is measured or commented. Also, the authors discuss nothing on the photorespiratory function of GS2 and its loss of function. Furthermore, I would like to know if the mutants grow well in soil or perlite/vermiculite soil, in nitrate or ammonium. Finally, I would like to know if the authors have tried to use other type of mutants as those mentioned above to prove the hypothesis.

- Regarding ammonium accumulation, the high contents at leaves of both *ami2* and *GLN2* mutant seem not to be toxic. However, this is quite surprising as high ammonium internal contents are currently considered quite toxic per se, and they are usually correlated to lower biomass during ammonium nutrition stress. This also should be commented.

- In the line 194 the authors mention that this “findings suggest that the *GLN2* reaction leads to excessive incorporation of ammonium-N into aminoacids in shoots when toxic levels of ammonium are present”. This idea is important, it is mentioned in other spots of the ms., and it must be demonstrated in this work. The assimilation of N could be derived to other molecules than aminoacids. I suggest you measure the contents of N (%) in leaves and roots of Col, *ami2* and GSN plants to demonstrate that higher N assimilation is taking place.

- Regarding the redaction of the ms., I consider that in the introduction section of the ms. It is insisted much in mutants as *GMP1* or *AMOS1* which are not relevant at a latter stage to comment on results or in the discussion section. In contrast, other ideas on ammonium assimilation as a tolerance mechanism, on the meaning of the displacement to lower C/N rates in aminoacids composition during ammonium nutrition or on the external acidification that ammonium entrance on the cell originates may be valuable to be commented in the introduction.

- On p19 the authors do not explain the exact conditions of the screening.

- The study lacks controls for ammonium delivery, such as ammonium chloride or ammonium sulphate (to unbalanced sulfate instead of chlorine).

Other minor questions:

- The supplementary table 1 can be presented as excel file.

- At the line 266, please remove “(25%)”. The percentage of ammonia that is used to increase the pH from 5.7 to 6.7 is not relevant and it should be mentioned in the material and methods.

- Line 84, I suggest to indicate “plastidic” localization of *GLN2* in the introduction

On the whole, this work might be valuable but the mentioned questions need to be addressed before it can be recommended for publication.

References

- Pérez-Delgado CM, García-Calderón M, Márquez AJ, Betti M (2015) Reassimilation of Photorespiratory Ammonium in Lotus japonicas Plants Deficient in Plastidic Glutamine Synthetase. PLoS ONE 10(6): e0130438. doi:10.1371/journal.pone.0130438
- Wallsgrave et al (1987) Barley Mutants Lacking Chloroplast Glutamine Synthetase—Biochemical and Genetic Analysis Plant Physiol. 83, 155-158 DOI: <https://doi.org/10.1104/pp.83.1.155>

The changes to the text within the file are highlighted using red text.

Reviewers' comments:

Reviewer #1 (Remarks to the Author):

Excessive ammonium assimilation by plastidic glutamine synthetase causes ammonium toxicity in *Arabidopsis thaliana*.

Screening *Arabidopsis* FOX lines under 10 mM toxic ammonium, authors identified over-expressing lines with higher tolerance to ammonium. Although the *ami2* line identified carries the 35S:GLN2 construct, this mutant was in fact defective for GLN2 probably due to silencing, and it was accordingly phenocopied by the *gln2* mutant. Then using elegant grafting assays, authors demonstrated that mutating GLN2 led to higher ammonium in shoots but lower tissue acidification, and then better tolerance to toxic ammonium levels. The fact that ammonium toxicity was due to acidification of shoot tissue was then verified using several approaches that combined grafting with other mutants and proton measurements. In addition, the study shows the antagonistic role of GLN2 in shoot and GLN1;2 in root regarding ammonium toxicity, and that *Nrt1.1* and GLN2 reduce ammonium tolerance by two different ways. There are many reports about ammonium toxicity and several mentioned acidification as one of the reason of the toxicity. The present study provides an elegant demonstration for that, and also reveals the surprising role of GLN2 in the tolerance to high ammonium. Indeed, one would have rather imagined that higher GLN2 activity leads to better tolerance to ammonium. This indeed challenge the textbook knowledge. This manuscript of high quality of science is also very well written and the results clearly presented are new and highly interesting for all the plant physiologists and molecular biologists interested in plant nutrition and nutrient signalling.

I will then have only few minor comments:

(Comment 1) From the paper of Rubio-Asensio, J.S. & Bloom, A.J., I would not say that high CO₂ inhibits nitrate utilization but reduce nitrate utilization, and it should also be said that even under high CO₂, nitrate utilization remains better for plant growth than ammonium utilization.

(Answer 1) According to the suggestion, we revised the manuscript (see Line 47, Line 450-451).

(Comment 2) It was described in barley (Kendall et al., 1986; Wallsgrove et al., 1987) and in Lotus japonicas (Orea et al., 2002) that GS2 mutants have photorespiratory symptoms when transferred from high CO₂ to air. Their hypersensitivity to air atmosphere was explained as due to lack of re-assimilation of photorespiratory ammonium. Is there any phenotype of gln2 mutant under nitrate conditions compared to WT? Maybe not on plates but at later stage of development in greenhouse? No role in photorespiration that necessitate high CO₂ for survival? This might be discussed.

(Answer 2) We determined the shoot growth of GLN2-deficient lines grown under 10 mM nitrate condition (see Fig. 1b, 2e and Supplementary Fig. S3d, S3f, S4d). Results show that the shoot fresh weights of GLN2-deficient lines were consistently lower than those of Col, which is the opposite trend of the shoot growth observed under ammonium.

*Recently, another research group reported that Arabidopsis GLN2-deficient mutants are able to complete their life cycle under photorespiratory conditions (~0.3% CO₂), although their shoots are smaller and slightly more chlorotic compared with the wild-type (Ferreira et al. 2019 *Plant Physiol. Biochem.* vol. 144, pp.365-374). They found that, even under conditions that minimize photorespiration (0.7% CO₂), the GLN2-deficient mutants show growth impairment, suggesting an important role of GLN2 in primary nitrogen assimilation. Also, we did not find a strong suppression of vegetative growth in *ami2* and *gln2* plants grown in the soil under photorespiratory conditions compared with the wild-type (see Supplementary Fig. S15a), although their seedling growth during 1 week after imbibition in the half-strength Murashige and Skoog media was significantly retarded (see Supplementary Fig. S15b,c). Collectively, Arabidopsis GLN2 would be functional for re-assimilation of photorespiratory ammonia but not essential for survival under photorespiratory conditions, probably because GLN2, GLN1s, and GDHs proteins could have redundant roles. The relevant description was added (see Line 422-442).*

(Comment 3) In Fig.4b the hierarchy clustering presents metabolite fold changes. FC means comparison of two conditions or two genotypes. Please explain in the legend what are exactly these FC values.

(Answer 3) The fold-change (FC) value on each amino acid was calculated by dividing the amino acid content of a sample by the mean content of all 18 samples. Log_2FC denotes the base-2 logarithm of FC (see Line 913-916).

(Comment 4) In Fig.11a, it looks that chloroplast structure of *gln2* and *ami2* is different from WT under ammonium conditions. Could you please comment?

(Answer 4) We agree with the Reviewer #1. The thylakoid membrane was swollen in *gln2*, but not in *ami2* under the nitrate condition (see Supplementary Fig. S13a), where the protein signal corresponding to GLN2 was slightly detected in *ami2* but not in *gln2* (Fig. 2c). The difference in residual amount of GLN2 might explain that in chloroplast structure.

The changes to the text within the file are highlighted using red text.

Reviewer #2 (Remarks to the Author):

(Comment 1) The article from Hachiya et al. aimed to identify the genes responsible for ammonium toxicity with full length overexpression lines. Surprisingly, a causal gene was glutamine synthetase 2, which was well characterized in barley. The physiological function of GS2 was defined in the assimilation of the ammonia during photorespiration in previous work. This work is ambitious because it challenges the textbook knowledge. However, the relation between GS2 and photorespiration was not discussed in this work. It is surprising because it is the textbook knowledge. This point is critical.

(Answer 1) We agree with the Reviewer #2. The mutants lacking GS2 in barley cannot survive under photorespiratory air conditions (Wallsgrove et al. 1987). In Lotus japonicus, the leaves of GS2-deficient mutants show severe necrotic phenotype when transferred from high to low CO₂ condition (Betti et al. 2014). Thus, it is widely believed that GS2 plays an essential role in re-assimilating photorespiratory ammonium. On the other hand, no mutant deficient in GLN2 has been isolated as a photorespiratory mutant from Arabidopsis plants, albeit vigorous screens has identified many genes responsible for photorespiration. A recent study reported that Arabidopsis GLN2-deficient mutants can complete their life cycle under photorespiratory air conditions, albeit smaller and slightly chlorotic than the wild-type (Ferreira et al. 2019). Interestingly, even under conditions that lower photorespiration, the GLN2-deficient mutants show growth impairment, suggesting an important role of GLN2 in primary nitrogen assimilation (Ferreira et al. 2019). Also, we did not find a strong suppression of vegetative growth of ami2 and gln2 plants grown in the soil under photorespiratory conditions compared with the wild-type (see Supplementary Fig. S15a), although their seedling growth during 1 week after imbibition in the half-strength Murashige and Skoog media was significantly retarded (see Supplementary Fig. S15b,c). Collectively, Arabidopsis GLN2 would be functional for re-assimilation of photorespiratory ammonia but not essential for survival under photorespiratory conditions, probably because GLN2, GLN1s, and GDHs proteins could have redundant roles. On the other hand, the present study did not clarify how improved ammonium insensitivity in Arabidopsis GLN2-deficient mutants is related to photorespiration. To elucidate this, it awaits many efforts in comparison between photorespiratory and non-photorespiratory conditions. The description about the relation between GS2 and photorespiration was added (see Line 422-442).

(Comment 2) I found some problems in data presentation and writing in this work. I hope my comments are valuable and helpful for this work to increase its value. The manuscript concludes that ammonium toxicity is acidic stress. It has been already known. Therefore, the novelty of this work should be that GS2 was a causative gene for acidic stress.

*(Answer 2) In this point, we do not agree with the Reviewer #2. To our best knowledge, it has not been clarified that acidic stress is a cause of ammonium toxicity, although it is widely known that ammonium as the sole N source consistently acidifies external media. Indeed, effects of ammonium application on intracellular pH differ depending on plant materials and external pH (Feng et al. 2020 J Exp Bot vol. 71, 4380-4392). We believe that, in *A. thaliana* shoots, the present study shows the first concrete evidence that acidic stress is one of the causes of ammonium toxicity.*

(Comment 3) I think this work showed well GS2 dependent proton production. However, the mechanism was still not clear. The manuscript explained that GS2 reaction release two proton in its reaction. I am not sure this point. Is there some other examples showing the strong impact on huge proton production by specific enzymatic reaction?

*(Answer 3) We agree with the Reviewer #2 in that GS2 reaction *per se* does not sufficiently explain the huge proton production. Unfortunately, we currently have few idea what enzymatic reaction contributes to the proton production, although hexokinase, phosphofructokinase, and glyceraldehyde phosphate dehydrogenase in the glycolysis and carbonic anhydrase are considered to be proton-producing pathways in cytosol (Sakano 1998 Plant Cell Physiol vol 39, 467-473). The comparison of metabolic flux between ammonium and nitrate as the sole N source with carbon isotope labeling may be useful for exploring the huge proton source. Thus, the expression about GS2-dependent proton production was weakened (see Line 361-362).*

*(Comment 4) According to the logic from the manuscript, *fd-gogat* mutants should be more sensitive to ammonium, and overexpression of *Fd-GOGAT* should be more tolerant to ammonium. I would suggest to probe them.*

*(Answer 4) We agree with the Reviewer #2. In order to further prove our model of ammonium toxicity, we would like to use these lines, although we cannot cultivate *GLU1*-deficient mutant due to a lack of high CO₂ system. In addition, we are now producing*

multiple mutants and overexpression lines on nitrogen assimilatory genes in the background of *GLN2*-deficient lines. We would like to report the follow-up in the near future.

(Comment 5) Ammonium supply changes the balance between GS2 and GS1. GS1 seems a major form in shoot in higher ammonium condition. Does GS1 contribute to proton production?

(Answer 5) We compared GS activity and proton concentrations among the shoots of Col, ami2, and gln1;2 gln1;3 (Supplementary Fig. S14c,d), because shoot expression of GLN1;2 and GLN1;3 encoding low-affinity GLN1 isozymes was significantly larger under 10 mM ammonium than 10 mM nitrate (Supplementary Fig. S3b). Results show that double knockout of GLN1;2/GLN1;3 did not significantly reduce the proton concentrations, whereas it decreased shoot GS activity by ca. 35%, being comparable to GLN2 deficiency. Thus, shoot GLN1;2 and GLN1;3 could assimilate excessive ammonium without causing proton accumulation in the shoot, albeit unknown mechanism. The description was added (see Line 405-421). On the other hand, we did not perform growth analysis using the plants grafted between Col and gln1;2 gln1;3. This is because Col and gln1;2 gln1;3 were not equally grown in our condition before grafting, preventing fair comparison of shoot growth.

(Comment 6) Since most of the data in this manuscript was derived from the two gs2 mutants. However, unfortunately both of them are mysterious. One was a overexpression line with suppression, the other was T-DNA insertion in 3'-UTR. Therefore, it is essential to discuss complementation lines. Recent publication shows successful GS2 overexpression in rice. Therefore, I suppose overexpression of GLN2 cDNA would always not result in co-suppression. Or, there is famous gs2 mutants in barley. It would be nice to test ammonium toxicity in barley mutant.

(Answer 6) In the revised manuscript, we observed an improvement in ammonium-tolerance in the 3rd GLN2-deficient line having a T-DNA at 10th exon (SALK_071292, named as gln2-2; see Supplementary Fig. S3a,e,f and Line 173-175, 464-465), which was recently isolated by another research group (Ferreira et al. 2019 Plant Physiol Biochem vol. 144, pp. 365-374). Ferreira et al. (2019) reported that the growth phenotype of SALK_071292 is similar to SALK_051953 (gln2 in our paper) and that both lines can

complete their life cycle under photorespiratory conditions. The enhanced ammonium tolerance in three independent *GLN2*-deficient lines (*ami2*, *gln2*, *gln2-2*) consistently indicates that *GLN2* causes ammonium toxicity. We did not check phenotypic complementation by transgene of *GLN2*, because, in general, it does not work in co-suppression lines. On the other hand, we produced overexpression lines using the coding sequence of *GLN2* under the control of 35S promoter, and two of homozygous T3 lines showed more than 20-fold expression of *GLN2*. Therefore, as indicated by the Reviewer #2, overexpression of *GLN2* is successful in *A. thaliana*. However, we did not find significant changes in total GS activity and ammonium tolerance in the overexpressors, implying post-translational regulation of *GLN2* protein.

The mutants lacking *GS2* in barley cannot survive under photorespiratory air conditions. In *Lotus japonicus*, the leaves of *GS2*-deficient mutants show severe necrotic phenotype when transferred from high to low CO₂ condition. On the other hand, Ferreira et al. (2019) reported that Arabidopsis *GLN2*-deficient mutants can complete their life cycle under air conditions, albeit smaller and slightly chlorotic than the wild-type. Also, we did not find a strong suppression of vegetative growth of *ami2* and *gln2* plants grown in the soil under photorespiratory conditions compared with the wild-type (see Supplementary Fig. S15a). These suggest that the physiological importance of *GS2* might differ among plant species. The comparison of *GS2*-deficient mutants in barley, *Lotus japonicus*, and other plant species are very interesting. However, we are currently not able to grow these mutants other than Arabidopsis ones, because we do not possess high CO₂ growth chamber to survive the mutants.

(Comment 7) This work discuss small seedlings in whole work. Since they are very small, the data may reflect the characteristics of seeds. Are the seed germination speed between Col and mutants same? Are the quality and quantity of nitrogen in Col and mutants similar?

(Answer 7) We did not find significant differences in total N and C concentrations in seeds and a time course of germination rate (Supplementary Fig. S15d-f and Line 434-436).

(Comment 8) This work often mixes different culture condition. It is confusing. Fig. 1a and Fig. 2d shows very different plant growth. The size of plant in Fig. 2d was much bigger than that in Fig. 1a. In Fig. 2d, plants were grown in pH 6.7 and were transferred to pH

5.7. The reason of pH switching was finally explained in Figure 6. The structure of data presentation is not easy to follow. In addition, phenotype was surely significant, but was not obvious in Fig. 2d. Even inside of Fig. 2 there was difference among b, c, and de. Plant growth and response to the ammonium should be very different in different culture condition as the manuscript claimed in introduction.

(Answer 8) The present study uses the plants at different developmental stages which were suitable for each analysis (including grafting and transfer experiments), but which would make it difficult for readers to interpret data, as indicated by the Reviewer #2. Therefore, in the revised manuscript, we showed the age of the plants as “x-d-old” in all the figure legends (see the legends). In addition, the reason why pH 6.7 was used for preculture of transfer experiment was described (see Line 482-485).

(Comment 9) This work shows fresh weight in whole work. It must be dry weight. Mutants might have much higher moisture than wild-type, and might be diluting ammonium.

(Answer 9) A lot of previous works have used the fresh weight as a measure of ammonium toxicity in *A. thaliana* (e.g. Hachiya et al. (2011) J Plant Res vol. 124, pp. 425-430; Li et al. (2012) Planta vol. 235, pp. 239-252; Li et al. (2012) Plant Physiol vol. 160, pp. 2040-2051; Sarasketa et al. (2014) J Exp Bot vol. 65, pp. 6023-6033; Jian et al. (2018) Plant Physiol vol. 178, pp. 1473-1488; Li et al. (2019) J Exp Bot vol. 70, pp. 1375-1388). Moreover, we checked that there was no significant difference in the ratio between fresh weight and dry weight among 12-d-old Col, *ami2*, and *gln2* 5 d after transfer to media 10 mM ammonium or 10 mM nitrate. Thus, in the present study, we used the fresh weight as a measure of ammonium toxicity.

(Comment 10) This work often shows ratio instead of absolute value. It masks some valuable information. It was difficult to understand the plant size difference between Fig. 1 and Fig. 2. It was not possible to compare the growth between Fig. 2d and Fig. 3c.

(Answer 10) According to the suggestion, we showed absolute fresh weights (see Figure 1d, 2e, 3c,d, 6a,c,e and Supplementary Figure S1b,c,e, S3d,f, S4a,c, S5c,d, S8, S12b, S15b).

(Comment 11) Fig. 2e, the data between ammonium and nitrate should be compared. Other Figures, 4a, 4c, 2b and Fig. 2e were different data interpretation.

(Answer 11) We compared the data between ammonium and nitrate by the Tukey-Kramer's multiple comparison test (see Fig. 2e).

(Comment 12) Fig. 4a, why ammonium concentration in ammonium grown plant and nitrate grown plant are similar?

(Answer 12) In Col shoots, the ammonium concentrations were $12.9 \pm 0.9 \mu\text{mol g}^{-1}$ under 10 mM ammonium and $2.2 \pm 0.4 \mu\text{mol g}^{-1}$ under 10 mM nitrate, respectively (Fig. 4a). The Welch's t-test detected the significant difference between them ($P = 0.0005989$), although a multiple comparison test did not find a significant difference between them, probably because of the extremely large ammonium concentrations in *ami2* and *gln2* shoots. Thus, the ammonium-grown wild-type had higher ammonium concentrations than the nitrate-grown plants. The description about ammonium concentration in Col shoots was added to the manuscript (see Line 210-212).

(Comment 13) Fig. 2, 3, 4 and 5, comparison between low ammonium and high ammonium should be discussed.

(Answer 13) In the present study, we mainly focused on the difference between high concentrations of ammonium and nitrate, as described in Introduction (see Line 52-57). The comparison between low ammonium and high ammonium was limited to the growth data in Figure 1d. In the revised manuscript, we added the description about the effects of low ammonium and high ammonium (Line 238-241).

(Comment 14) L24 Plant can use organic nitrogen.

(Answer 14) We agree with the Reviewer #2. The description was added (see Line 24).

(Comment 15) L61, there are several definitions of ammonium toxicity in different ammonium concentration in previous works. This work should compare previous definitions and define its ammonium toxicity in the manuscript.

(Answer 15) According to the suggestion, we described the definition in the revised manuscript (see Line 52-57).

(Comment 16) L375-377, the isolation of mutants should be explained in detail.

How are they isolated, and how are they confirmed as mutants?

(Answer 16) We detailed the procedure of the screening (see Line 492-508).

(Comment 17) L392-393, this medium contains MES and sucrose?

The medium used in this work was not clear. Why MES was not able to buffer the acidification in this work?

(Answer 17) The information about the pH 6.7 medium was added (see Line 482-485). MES can buffer the pH of media, but does not work to maintain the pH of plant tissues/cells, because it is considered that Good's buffer including MES cannot pass through the biological membrane. In addition, MES was not included in the incubation liquid media for determination of H⁺ concentration in a water extract of shoots and H⁺ efflux from shoots (see Line 633 and 647), allowing the detection of pH changes.

The changes to the text within the file are highlighted using red text.

Reviewer #3 (Remarks to the Author):

This work presents interesting data on the glutamine synthetase-2 (GS2) assimilation reaction as responsible, through the H⁺ release, of an acidic stress which would be subjacent to the stress by ammonium nutrition. The authors provide a radical new idea on the underlying mechanisms of stress by ammonium, which is studied only in the plant *Arabidopsis*. The work includes an extensive set of data around the biochemical mechanisms, the gene expression and metabolite induction, which support their hypothesis. The data are also supported by an important set of supplementary figures and a table. In general, the data seems well presented although some questions need to be further clarified.

(Comment 1) The role of GS2 in Arabidopsis during photorespiration may be the most difficult point to accept from this new hypothesis. It is considered among plant biologist that GS2 has an essential role on the photorespiration during recycling of the photorespiratory ammonium and in the assimilation of ammonium from reduction of nitrate at the plastids. Thus, the barley mutants on GS2 needed to be growth on 1% oxygen to avoid growth limitation during photorespiratory conditions (Wallsgrave et al., 1987). In the Lotus japonicum GS2 mutants, the plants needed to be cultivated at least at 4000 ppm of CO₂, while 7000 ppm of CO₂ were regularly used to avoid photorespiratory conditions (Perez-Delgado et al., 2015). In this work, the GS2 expression is 95% lost in ami2 mutant according to the authors, and beyond that in GSN mutant. This should at least induce a phenotype of the ami2 and GSN plants under nitrate but nothing is measured or commented. Also, the authors discuss nothing on the photorespiratory function of GS2 and its lost of function. Furthermore, I would like to know if the mutants grow well in soil or perlite/vermiculite soil, in nitrate or ammonium.

(Answer 1) We determined the shoot growth of GLN2-deficient lines grown under 10 mM nitrate condition (see Fig. 1b, 2e and Supplementary Fig. S3d, S3f, S4d). Results show that the shoot fresh weights of GLN2-deficient lines were consistently lower than those of Col, which is the opposite trend of the shoot growth observed under ammonium.

The mutants lacking GS2 in barley cannot survive under photorespiratory air conditions (Wallsgrave et al. 1987). In Lotus japonicus, the leaves of GS2-deficient mutants show severe necrotic phenotype when transferred from high to low CO₂

condition (Betti et al. 2014). Thus, it is widely believed that GS2 plays an essential role in re-assimilating photorespiratory ammonium. On the other hand, no mutant deficient in *GLN2* has been isolated as a photorespiratory mutant from Arabidopsis plants, albeit vigorous screens has identified many genes responsible for photorespiration. A recent study reported that Arabidopsis *GLN2*-deficient mutants can complete their life cycle under photorespiratory air conditions, albeit smaller and slightly chlorotic than the wild-type (Ferreira et al. 2019). Interestingly, even under conditions that lower photorespiration, the *GLN2*-deficient mutants show growth impairment, suggesting an important role of *GLN2* in primary nitrogen assimilation (Ferreira et al. 2019). Also, we did not find a strong suppression of vegetative growth of *ami2* and *gln2* plants grown in the soil under photorespiratory conditions compared with the wild-type (see Supplementary Fig. S15a), although their seedling growth during 1 week after imbibition in the half-strength Murashige and Skoog media was significantly retarded (see Supplementary Fig. S15b,c). Collectively, Arabidopsis *GLN2* would be functional for re-assimilation of photorespiratory ammonia but not essential for survival under photorespiratory conditions, probably because *GLN2*, *GLN1s*, and *GDHs* proteins could have redundant roles. On the other hand, the present study did not clarify how improved ammonium insensitivity in Arabidopsis *GLN2*-deficient mutants is related to photorespiration. To elucidate this, it awaits many efforts in comparison between photorespiratory and non-photorespiratory conditions. The description about the relation between *GS2* and photorespiration was added (see Line 422-442).

(Comment 2) Finally, I would like to know if the authors have tried to use other type of mutants as those mentioned above to prove the hypothesis.

(Answer 2) In the revised manuscript, we observed an improvement in ammonium-tolerance in the 3rd *GLN2*-deficient line having a T-DNA at 10th exon (SALK_071292, named as *gln2-2*; see Supplementary Fig. S3a,e,f and Line 173-175, 464-465), which was recently isolated by another research group (Ferreira et al. 2019 Plant Physiol Biochem vol. 144, pp. 365-374). Ferreira et al. (2019) reported that the growth phenotype of SALK_071292 is similar to SALK_051953 (*gln2* in our paper) and that both lines can complete their life cycle under photorespiratory conditions. The enhanced ammonium tolerance in three independent *GLN2*-deficient lines (*ami2*, *gln2*, *gln2-2*) consistently indicates that *GLN2* causes ammonium toxicity. We did not check phenotypic

complementation by transgene of *GLN2*, because, in general, it does not work in co-suppression lines. On the other hand, we produced overexpression lines using the coding sequence of *GLN2* under the control of 35S promotor, and two of homozygous T3 lines showed more than 20-fold expression of *GLN2*. Therefore, as indicated by the Reviewer #2, overexpression of *GLN2* is successful in *A. thaliana*. However, we did not find significant changes in total GS activity and ammonium tolerance in the overexpressors, implying post-translational regulation of *GLN2* protein. In order to further prove our model of ammonium toxicity, we are now producing multiple mutants and overexpression lines on nitrogen assimilatory genes in the background of *GLN2*-deficient lines. We would like to report the follow-up in the near future.

(Comment 3) - Regarding ammonium accumulation, the high contents at leaves of both ami2 and GLN2 mutant seem not to be toxic. However, this is quite surprising as high ammonium internal contents are currently considered quite toxic per se, and they are usually correlated to lower biomass during ammonium nutrition stress. This also should be commented.

(Answer 3) We were also surprised to observe remarkable ammonium accumulation with shoot growth enhancement in GLN2-deficient lines. On the other hand, we have previously demonstrated that nitrate addition at adequate concentrations mitigates ammonium toxicity without reducing ammonium accumulation (Hachiya et al. 2012 Plant Cell Physiol vol. 53, 577-591). Moreover, Barth et al. (2010) reported that high levels of ammonium (approx. 25 $\mu\text{mol g}^{-1}$ fresh weight) were accumulated in Col seedlings grown under normal Murashige and Skoog medium, where seedlings show healthy growth (Barth et al. 2010 J Exp Bot vol. 61, 379-394). It should be noted that Murashige and Skoog medium includes approx. 20 mM ammonium. Thus, ammonium accumulation per se in the plant tissue might be not so toxic for plants, although the cellular compartment of ammonium accumulation is undetermined in the present study.

(Comment 4) - In the line 194 the authors mention that this "findings suggest that the GLN2 reaction leads to excessive incorporation of ammonium-N into aminoacids in shoots when toxic levels of ammonium are present". This idea is important, it is mention in other spots of the ms., and it must be demonstrated in this work. The assimilation of N could be derived to other molecules than amino acids. I suggest you

measure the contents of N (%) in leaves and roots of Col, ami2 and GSN plants to demonstrate that higher N assimilation is taking place.

(Answer 4) We found that total N and protein concentrations were generally larger in ammonium-grown shoots than nitrate-grown shoots (see Supplementary Fig. S10a,b). In ammonium-grown shoots, the total N concentrations were significantly reduced by *GLN2* deficiency (see Supplementary Fig. S10a), and the protein concentrations were marginally decreased (see Supplementary Fig. S10b). Collectively, we conclude that the shoot *GLN2* would facilitates accumulation of assimilated N in the shoot under toxic ammonium condition. The relevant descriptions were also added (see Line 228-234). We did not determine root concentrations of total N and protein, because the root penetration into solidified medium prevented accurate determination of root fresh weights.

(Comment 5) -Regarding the redaction of the ms., I consider that in the introduction section of the ms. It is insisted much in mutants as GMP1 or AMOS1 which are not relevant at a latter stage to comment on results or in the discussion section. In contrast, other ideas on ammonium assimilation as a tolerance mechanism, on the meaning of the displacement to lower C/N rates in aminoacids composition during ammonium nutrition or on the external acidification that ammonium entrance on the cell originates may be valuable to be commented in the introduction.

(Answer 5) Thank you very much for your suggestion. However, we regret that we were not able to reconstruct the Introduction in the revised manuscript successfully.

(Comment 6) - On p19 the authors do not explain the exact conditions of the screening.

(Answer 6) We detailed the procedure of the screening (see Line 492-508).

(Comment 7) - The study lacks controls for ammonium delivery, such as ammonium chloride or ammonium sulphate (to unbalanced sulfite instead of chlorine).

(Answer 7) We confirmed that a similar growth enhancement in *ami2* was observed in the media containing 10 mM NH₄Cl as the sole N source (see Supplementary Fig. S1a,b and Line 119-122).

Other minor questions:

(Comment 8) -The supplementary table 1 can be presented as excel file.

(Answer 8) We agree with the Reviewer #3. We will present the Supplementary Tables as excel file for readers' convenience.

(Comment 9) - At the line 266, please remove "(25%)". The percentage of ammonia that is used to increase the pH from 5.7 to 6.7 is not relevant and it should be mention in the material and methods.

(Answer 9) We agree with the Reviewer #3. We deleted "25%" from the revised manuscript (see Line 315). "25% (v/v)" was added in the Figure legend (see Line 948).

(Comment 10) - Lane 84, I suggest to indicate "plastidic" localization of GLN2 in the introduction

(Answer 10) We added the information in the revised manuscript (see Line 29, 87).

On the whole, this work might be valuable but the mentioned questions need to be addressed before it can be recommended for publication.

References

Pérez-Delgado CM, García-Calderón M, Márquez AJ, Betti M (2015) Reassimilation of Photorespiratory Ammonium in Lotus japonicas Plants Deficient in Plastidic Glutamine Synthetase. PLoS ONE 10(6): e0130438. doi:10.1371/journal.pone.0130438

Wallsgrave et al (1987) Barley Mutants Lacking Chloroplast Glutamine Synthetase—Biochemical and Genetic Analysis Plant Physiol. 83, 155-158
DOI: <https://doi.org/10.1104/pp.83.1.155>

REVIEWERS' COMMENTS

Reviewer #1 (Remarks to the Author):

All the concerns raised in my report have been taken into account and I consider that authors' responses are satisfactory.

C. Masclaux-Daubresse

Reviewer #2 (Remarks to the Author):

The authors successfully addressed all the points I have proposed based on their previous version. I have no more suggestions and questions.

Reviewer #3 (Remarks to the Author):

Regarding the manuscript by Hachiya et al, I found all the responses from authors satisfactory. I consider that the ms has greatly improved and I would like to recommend it for publication. The articles is of great novelty and represents and a lot of work. Also, it is true that much work remain to be done to fully understand the implications as well as details of the mechanism.

REVIEWERS' COMMENTS

Comments:

Reviewer #1 (Remarks to the Author):

All the concerns raised in my report have been taken into account and I consider that authors' responses are satisfactory.

C. Masclaux-Daubresse

Reviewer #2 (Remarks to the Author):

The authors successfully addressed all the points I have proposed based on their previous version. I have no more suggestions and questions.

Reviewer #3 (Remarks to the Author):

Regarding the manuscript by Hachiya et al, I found all the responses from authors satisfactory. I consider that the ms has greatly improved and I would like to recommend it for publication. The article is of great novelty and represents a lot of work. Also, it is true that much work remains to be done to fully understand the implications as well as details of the mechanism.

Answer: We truly appreciate the effort that the reviewers have dedicated to providing constructive and valuable feedback to significantly improve our manuscript. In the near future, we will present further information to understand the underlying mechanism.